# CoRe: Combined Rewards with Vision-Language Model Feedback for Preference-Aligned Reinforcement Learning

Hexian Ni [1 2]   Tao Lu [1]   Yinghao Cai [1]

## Abstract

Reward design remains a central challenge in reinforcement learning (RL). Hand-crafted rewards are often difficult to specify and may lead to suboptimal policies, while learned rewards from preferences can suffer from inefficiency and unstable training. Inspired by the dual nature of human learning explored in cognitive science, we decompose rewards into two complementary components: Formal Rewards (FR), explicitly designed based on task knowledge, and Residual Rewards (RR), learned from observations to capture implicit and nuanced preferences. Based on this decomposition, we propose CoRe, a hybrid framework that integrates FR and RR with vision-language models (VLMs) feedback to achieve preference-aligned policies without human involvement. Our contributions are twofold: (1) We propose a Formal Reward Module (FRM) that leverages VLMs to iteratively design and optimize FR based on task knowledge and preference feedback, enabling the continual improvement of policy during training; (2) We introduce a Residual Reward Module (RRM) that learns RR from video-level preference by employing VLMs to generate preference labels and capturing nuanced rewards that complement FR, ensuring alignment with human intent. Through the synergy of FRM and RRM, CoRe enables the automatic construction of reliable rewards that are efficient and preference-aligned. Extensive experiments demonstrate that CoRe outperforms existing approaches in terms of policy learning effectiveness and efficiency on ten robotic manipulation tasks in simulation and five real-

worlds. Videos can be found on our project website: https://core-2026.github.io/

## 1. Introduction

Reinforcement learning (RL) has achieved notable success in gaming AI (Mnih et al., 2013; Silver et al., 2016; Berner et al., 2019), autonomous driving (Shalev-Shwartz et al., 2016; Sallab et al., 2017; Kiran et al., 2021), and robotic manipulation (Kober et al., 2013; Kalashnikov et al., 2018; Geng et al., 2022), etc. However, a central challenge of RL lies in designing reward functions which serve as the learning signals for policy optimization. Designing appropriate rewards is notoriously difficult: poorly specified rewards often lead to suboptimal or unsafe behaviors, while shaping rewards demands substantial domain expertise and engineering effort. To address this issue, recent research has explored alternative approaches that leverage human demonstrations (Ho & Ermon, 2016; Christiano et al., 2017; Eteke et al., 2020) or large-scale pre-trained models (Sontakke et al., 2023; Ma et al., 2024; Wang et al., 2024; Xie et al., 2024; Luu et al., 2025) to automatically derive rewards, thereby reducing reliance on hand-crafted reward engineering.

A prominent direction is preference-based reinforcement learning (PbRL) (Lee et al., 2021a; Liu et al., 2022; Liang et al., 2022; Park et al., 2022; Cheng et al., 2024), where reward models are derived from human comparisons of trajectory segments. It eliminates the need for hand-crafted rewards, and aligning with human preferences effectively avoids reward hacking (Amodei et al., 2016), achieving policies that meet human expectations. However, this paradigm often requires a large number of preference labels, which is costly and limits scalability. To mitigate this burden, recent work has explored leveraging large language models (LLMs) and vision-language models (VLMs) to directly generate or synthesize preference labels based on textual or visual goals (Klissarov et al., 2024; Wang et al., 2024; Venkataraman et al., 2025; Lin et al., 2024b; Tu et al., 2025; Wang et al., 2025; Luu et al., 2025). Another line of research exploits the code-generation and reasoning capabilities of LLMs to output executable reward function code, thereby avoiding intricate reward engineering (Yu et al., 2023; Ma

[1]State Key Laboratory of Multimodal Artificial Intelligence Systems, Institute of Automation, Chinese Academy of Sciences, Beijing, China [2]School of Artificial Intelligence, University of Chinese Academy of Sciences, Beijing, China. Correspondence to: Tao Lu <tao.lu@ia.ac.cn>.

*Proceedings of the 43rd International Conference on Machine Learning*, Seoul, South Korea. PMLR 306, 2026. Copyright 2026 by the author(s).

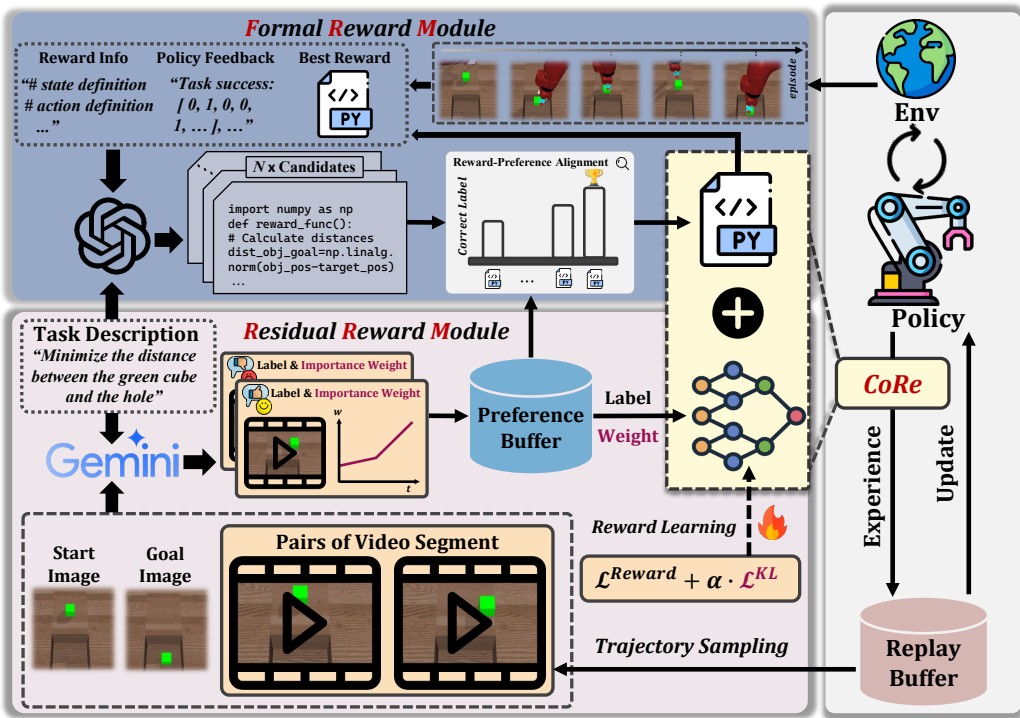

*Figure 1.* Illustration of CoRe. CoRe decomposes the RL reward into a formal reward (FRM) and a residual reward (RRM). FRM leverages LLMs together with VLM-based preferences to iteratively generate and refine code-based rewards, facilitating effective RRM exploration, high-quality preference labeling, and stable training. RRM incorporates video-level preferences and state-importance from VLMs to complement FRM and ensure alignment with human intent. By integrating these complementary signals, CoRe enables feedback-efficient and preference-aligned policy learning without human involvement.

et al., 2024; Zeng et al., 2024a; Li et al., 2024; Xie et al., 2024). In addition, VLMs such as CLIP (Radford et al., 2021) have also been employed to score states based on the similarity of visual or textual task objectives to obtain task rewards at each step (Cui et al., 2022; Mahmoudieh et al., 2022; Sontakke et al., 2023; Ma et al., 2023; Rocamonde et al., 2024; Adeniji et al., 2023). Collectively, these approaches reduce dependence on manual reward design and broaden the applicability of RL. Despite these advances, significant limitations remain. Learned reward models, such as those inferred from human preference, often suffer from low feedback efficiency (Lee et al., 2021a; Liang et al., 2022; Verma & Kambhampati, 2023), unstable training and poor generalization to unseen scenarios. Reward functions generated by LLMs or VLMs tend to be coarse or noisy (Yu et al., 2023; Li et al., 2024; Zeng et al., 2024b), and may fail to capture the fine-grained structure required for complex manipulation tasks (Ma et al., 2024; Xie et al., 2024; Chen & Gombolay, 2025). A key open problem, therefore, is how to obtain reward signals that are both feedback-efficient and reliably aligned with task-specific human intentions.

Cognitive science research (Dong et al., 2020; Schneider & Simonsmeier, 2025; Bittermann et al., 2023) shows that human learning is influenced by prior knowledge, which

supports encoding, comprehension, and schema construction processes in novel tasks. For aspects of tasks involving specific dynamics, constraints, or feedback, it often requires further iterative refinement through interaction with the environment. Motivated by this, we decompose rewards into two complementary components: (i) Formal Rewards (FR), which is explicitly designed and encoded based on task knowledge, representing the part that can be directly specified; and (ii) Residual Rewards (RR), which captures task aspects that are difficult to hand-design and must be learned from observations, enabling adaptation to nuanced preferences that cannot be easily encoded. This decomposition allows task rewards to combine structured and task-relevant features with flexible preference alignment, reducing the learning difficulty of preference-based RL. Then in this paper, we present CoRe, a hybrid framework that integrates FR and RR with vision-language models (VLMs) feedback to achieve preference-aligned policies in robotic manipulation tasks. Our contributions are twofold: (1) We develop a Formal Reward Module (FRM) that leverages VLM-based preference to automatically design and optimize formal rewards, guiding continuous policy improvement while maintaining alignment with human intentions. (2) We introduce a Residual Reward Module (RRM) that learns residual rewards from video-level preferences. By employing VLMs to gen-

erate preference labels and assess state importance, RRM captures fine-grained features of manipulation tasks and complements formal ones. Through the synergy between FRM and RRM, CoRe effectively leverages the common-sense knowledge from VLMs which is aligned with human preference and enables the automatic construction of reliable, preference-aligned rewards, eliminating the need for manual reward engineering. Extensive experiments on both simulated and real-world manipulation tasks demonstrate that CoRe outperforms existing methods in terms of effectiveness and efficiency.

## 2. Related Work

**LLMs/VLMs for Reward Design in RL.** Leveraging the strong code generation, in-context understanding, and instruction-following capabilities of coding-oriented LLMs, recent work has explored using LLMs to directly generate RL reward functions, thereby avoiding complex reward engineering (Yu et al., 2023; Ma et al., 2024; Xie et al., 2024; Ryu et al., 2025; Li et al., 2024; Zeng et al., 2024b; Huang et al., 2025). Ma et al. (2024) introduces an automated reward design framework powered by GPT-4 (OpenAI, 2025). Without requiring detailed prompt engineering, it can generate human-level reward functions based on the environment code through evolutionary search and reward reflection. Huang et al. (2025) presents a reward-policy co-evolution framework that avoids retraining policies from scratch for each reward candidate. Zeng et al. (2024b) proposes to parameterize rewards using LLMs and minimize the discrepancy between predicted rewards and the LLM's ranking over trajectory segments, which stabilizes reward learning and improves policy consistency. To better align with human intent, some studies have moved beyond textual task descriptions and instead utilize visual demonstrations to guide reward design (Zeng et al., 2024a; Chen & Gombolay, 2025; Stamatopoulou et al., 2025; Mahesheka et al., 2024; Ye et al., 2025). Zeng et al. (2024a) transforms video demonstrations into textual descriptions of key events, which are then interpreted by LLMs to generate evaluation benchmarks. This enables accurate reward modeling and policy learning directly from videos. Chen & Gombolay (2025) leverages VLMs to extract visual feature functions from video demonstrations and task prompts, then applies inverse reinforcement learning to recover task-aligned rewards. This approach generalizes well to out-of-distribution tasks. However, these code-generation methods often require access to the entire environment code to design detailed reward functions (Wang et al., 2024; Tu et al., 2025; Luu et al., 2025). In this paper, we employ LLMs to refine the reward function and utilize preference labels for reward search. This ensures the reward function remains continuously aligned with preference, enabling incremental performance improvements of the policy. We also do not require environment code to obtain fine-grained rewards due to the design of the learnable residual reward.

**Reward Learning from Preference.** PbRL provides a solution to avoid reward engineering (Christiano et al., 2017; Lee et al., 2021b;a; Park et al., 2022; Liu et al., 2022; Liang et al., 2022; Cheng et al., 2024). However, for complex manipulation tasks, PbRL typically requires a large number of preference labels to learn accurate reward models, which can be costly when relying on human labeling. To address this challenge, several works have proposed informative query selection schemes that identify high-value samples to accelerate reward learning (Christiano et al., 2017; Lee et al., 2021a; Hu et al., 2024). Another line of work aims to optimize the reward learning framework (Liu et al., 2022; Liang et al., 2022; Liu et al., 2023; Verma & Metcalf, 2024). Liu et al. (2022) proposed Meta-Reward-Net (MRN) which integrates Q-function performance into reward learning, enabling efficient policy optimization with limited labels. Verma & Metcalf (2024) leverages a learned world model to estimate the importance of individual states within a trajectory segment, guiding rewards to be proportional to these importance scores, thus accelerating reward and policy learning. These focus on enhancing feedback efficiency, thereby reducing the number of required labels. Recently, LLMs/VLMs have shown impressive capabilities in common sense reasoning and visual understanding (OpenAI, 2025; Comanici et al., 2025). Motivated by this, several studies have explored replacing human labeling with LLMs/VLMs to provide preference labels automatically (Wang et al., 2024; 2025; Tu et al., 2025; Venkataraman et al., 2025; Liu et al., 2025; Ghosh et al., 2025; Luu et al., 2025). Wang et al. (2024) directly queries VLMs for image-level preference labels using goal text, avoiding reward noise introduced by direct VLMs scoring (Cui et al., 2022; Mahmoudieh et al., 2022; Sontakke et al., 2023; Ma et al., 2023; Rocamonde et al., 2024; Adeniji et al., 2023). Wang et al. (2025) sample multiple LLM-based evaluation functions to generate score-level synthetic preference labels, achieving performance comparable to expert-scripted teachers. In this paper, we propose using VLMs to capture video-level preference and evaluate state importance, enabling efficient reward learning. Our method can significantly enhance feedback efficiency and the performance of reward models.

## 3. Method

In this section, we introduce our CoRe, a framework that combines the Formal Reward Module (FRM) and the Residual Reward Module (RRM) with VLMs feedback to achieve preference-aligned RL without human involvement.

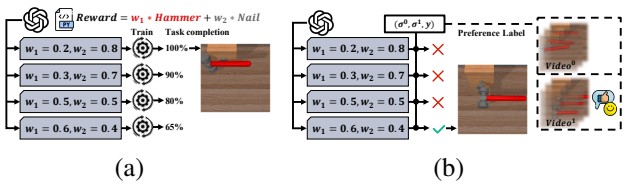

(a)          (b)

*Figure 2.* (a) Relying on task completion to evaluate reward candidates incurs additional training and may lead to reward exploitation (hammer handle striking). (b) Reward-Preference Alignment evaluates candidates using preference labels, constraining them to align with preference and ensuring the policy meets human intent.

### 3.1. Formal Reward Module (FRM)

Code-based formal rewards defined over explicit task features are easy to design and can efficiently express simple task goals. Recent work shows that LLMs can automatically generate and iteratively refine executable formal reward code (Ma et al., 2024). At iteration $n$, given the task description $t_d$, the environment code (we only need some task knowledge $t_s$, such as the definition of states, not code), the previous best reward $\hat{r}_{best}^{n-1}$ and the policy feedback $t_f^{n-1}$ collected under that reward, LLMs generate a set of reward candidates $\hat{r}^n = [\hat{r}_1^n, \hat{r}_2^n, ..., \hat{r}_K^n]$. The task completion function $F_c(\hat{r}_k^n)$ evaluates the policy trained with candidate $\hat{r}_k^n$. The best reward is selected accordingly:

$$\hat{r}_{best}^n = \max_{\hat{r}_k^n \in \hat{r}^n} F_c(\hat{r}_k^n). \tag{1}$$

However, this approach may not be sufficient for complicated or abstract task goals, such as aligning with human expectations. As shown in Figure 2a, LLMs identify the Hammer task features of formal rewards as "Hammer" and "Nail," representing the hammering action and the nail's location, respectively. According to Equation (1), the reward candidate with high "Nail" weights will be selected as optimizing the "Nail" reward component directly yields the highest task completion. Nevertheless, excessive focus on task completion may lead to undesirable behavior, such as striking with the hammer handle. To overcome this issue, we replace task completion-based evaluation with preference-guided reward search, where the reward candidate most consistent with preference is selected (Figure 2b). We denote this process as Reward-Preference Alignment.

Specifically, there exists a preference dataset $\mathcal{D}$ containing triples $(\sigma^0, \sigma^1, y)$. Each trajectory segment $\sigma = \{(s_k, a_k), \dots, (s_{k+H}, a_{k+H})\}$ represents a short part of an episode in a standard Markov Decision Process (Sutton & Barto, 2018), where $s_t$ and $a_t$ denote the state and action at time step $t$, and $H$ is the segment length. The label $y \in \{(0,1), (1,0), (0.5, 0.5)\}$ indicates preference for $\sigma^1$, preference for $\sigma^0$, or no preference, respectively. $F_p(\hat{r}_k^n, \mathcal{D})$ represents the preference prediction accuracy rate of $\hat{r}_k^n$ for

$\mathcal{D}$, and the best reward $\hat{r}_{best}^n \in \hat{r}^n$ can be defined as:

$$\hat{r}_{best}^n = \max_{\hat{r}_k^n \in \hat{r}^n} F_p(\hat{r}_k^n, \mathcal{D}), \tag{2}$$

$$F_p(\hat{r}_k^n, \mathcal{D}) = \mathbb{E}_{(\sigma^0, \sigma^1, y) \sim \mathcal{D}} \left[ y(0) * \mathbb{I}(\hat{R}_k^n(\sigma^0) > \hat{R}_k^n(\sigma^1)) \right. \\ \left. + y(1) * \mathbb{I}(\hat{R}_k^n(\sigma^1) > \hat{R}_k^n(\sigma^0)) \right], \tag{3}$$

where $\hat{R}_k^n(\sigma^i) = \sum_t \hat{r}_k^n(s_t^i, a_t^i)$. This method enables formal rewards to be gradually optimized and aligned with human preferences, ensuring continuous policy improvement. It can also quickly verify the reward candidate, avoid training and evaluation from scratch, and improve the efficiency of policy learning. Importantly, it dynamically aligns the FRM's optimization objective with the preference to ensure stability throughout the training process.

### 3.2. Residual Reward Module (RRM)

Nonlinear reward models with a large number of parameters excel in expressing complex and abstract task goals, but often suffer from feedback inefficiency (Wirth et al., 2017; Ghosh et al., 2025; Wang et al., 2024). This inefficiency arises because the single state-based preference conveys limited information, while the segment-based preference (e.g., trajectory preference) makes accurate state-wise credit assignment difficult. To address these issues, we propose video-level preference image-based reward learning from VLMs' feedback. Rather than relying on low-expression preference, such as natural language (Klissarov et al., 2024; Lin et al., 2024b), coordinate-based feedback (Tu et al., 2025), or single-frame image (Wang et al., 2024), which may not adequately express preference, videos can better capture task dynamics and convey richer information. For credit assignment, we introduce a two-stage multimodal prompting query based on the chain-of-thought (Wei et al., 2022) to assess the relative importance of individual image frame which serves as an approximate prior for rewards.

Specifically, the segment with image observations $I$ is formed as $\sigma = \{(s_k, a_k, I_k), \dots, (s_{k+H}, a_{k+H}, I_{k+H})\}$. To enhance visual understanding, we input task description $l_d$, start image $I_s$, goal image $I_d$ and videos $\{I_t^i\}_{i \in \{0,1\}}$ into VLMs $F_{VLM}$, and obtain the preference label:

$$y = F_{VLM}\left(l_d, I_s, I_d, \{I_t^i\}_{i \in \{0,1\}}\right). \tag{4}$$

When trajectory segments are not meaningfully comparable, VLMs output "incomparable" labels, which are excluded from residual reward training. During preference labeling, VLMs also assess task completion across the frames, producing the per-frame importance weight over the full segment $W^i = \{w_t^i\}_{i \in \{0,1\}}$, where $w_t^i$ denotes the state importance of frame $t$ in $\sigma^i$. Each preference feedback is stored as a

triple $(\sigma^0, \sigma^1, y, W^0, W^1)$ in the preference data $\mathcal{D}$. Based on this, we use the estimated importance weight as a prior over predicted state rewards. In addition to optimizing the cross-entropy loss $\mathcal{L}^{Reward}$ to match preference (Lee et al., 2021a), we enforce consistency between the predicted state reward distribution and the prior importance weight via the Kullback–Leibler divergence (Csiszár, 1975). The resulting objective can be defined as:

$$\mathcal{L} = \mathcal{L}^{Reward} + \alpha \cdot \mathcal{L}^{KL}, \tag{5}$$

$$\mathcal{L}^{KL} = \underset{(\sigma^0, \sigma^1, W^0, W^1) \sim \mathcal{D}}{\mathbb{E}} \left[ D_{KL}(W^0 \| R_\psi^0) \\ + D_{KL}(W^1 \| R_\psi^1) \right], \tag{6}$$

where $R_\psi^i = \left[ \hat{r}_\psi(I_k^i), ..., \hat{r}_\psi(I_{k+H}^i) \right]$. $\alpha$ is a weighting coefficient. $\hat{r}_\psi$ is the reward model. Video-level preference reward learning enables the residual reward to capture VLM-aligned preference effectively. This significantly accelerates reward learning and improves feedback efficiency.

## 4. Experiments

### 4.1. Experimental Setup

**Tasks.** We evaluate CoRe on a diverse set of robotic manipulation tasks involving *rigid*, *articulated*, and *deformable* objects. These tasks span a wide range of primitive manipulation skills, including pushing, pulling, pressing, twisting, and grasping. Specifically, we consider seven from *MetaWorld* (Yu et al., 2020) (Soccer, Sweep Into, Drawer Open, Button Press, Dial Turn, Hammer and Peg Insert) and three from *SoftGym* (Lin et al., 2021) (Fold Cloth, Straighten Rope and Pass Water) which cover both rigid-body and deformable-object manipulation. Detailed task descriptions and visualizations are provided in the Appendix A.

**Baselines.** We compare CoRe against representative methods that leverage pretrained LLMs or VLMs for reward construction, including: (i) *similarity-based reward scoring*: CLIP Score (Rocamonde et al., 2024) (ii) *code-based reward generation*: Eureka (Ma et al., 2024) and Text2Rward (Xie et al., 2024) and (iii) *preference- or rating-based reward learning from visual feedback*: RL-VLM-F (Wang et al., 2024), PrefVLM (Ghosh et al., 2025) and ERL-VLM (Luu et al., 2025). In addition, we report results using *Environment Sparse Reward (Env Sparse)* and *Environment Dense Reward (Env Dense)* provided by the benchmark. Detailed description can be found in the Appendix C.

**Implementation Details.** We adopt SAC (Haarnoja et al., 2018) for policy learning with state observations. CLIP Score, RL-VLM-F, PrefVLM, and ERL-VLM follow PEBBLE's unsupervised pre-training (Lee et al., 2021a), while other methods train policies directly. To ensure fairness, all

methods have the same hyperparameters for policy learning, with the only difference being the reward. Reward functions for FRM, Eureka and Text2Reward are generated by GPT-4.1 mini (OpenAI, 2025), with five reward searches performed, sampling four reward candidates each time. The reward feedback for Text2Reward is provided by the author. PrefVLM uses LIV (Ma et al., 2023) as the VLM. Preference labels for RRM, RL-VLM-F and ERL-VLM are provided by Gemini 2.0 Flash (Comanici et al., 2025) with a uniform sampling scheme for queries. The number of labels and images is shown in Table 3. The reward model is trained on images rendered by the simulator. To reduce computational cost and improve query efficiency in RRM, each video is downsampled before being input into VLM. The importance weight evaluated by VLM is first linearly interpolated and then normalized using a softmax function to obtain the importance weight of the full segment. To standardize reward scales, FRM and RRM rewards are clipped to $[-0.5, 0.5]$. The total reward is the sum of FRM and RRM. The full procedure for CoRe in Algorithm 1. Detailed hyperparameters and training implementation for all methods can be found in Appendix B.

To ensure a fair comparison among all methods, we render only task-relevant objects during training of the image-based reward model by making the robot arm (MetaWorld) and pickers (SoftGym) transparent and adjusting the camera view to focus on the objects. We evaluate performance using the success rate for MetaWorld and the episode return for SoftGym. Metrics are recorded per episode and averaged over three runs.

### 4.2. Main Results

Figure 3 shows the learning curves across all tasks. CoRe demonstrated faster learning than humans in Sweep Into and Hammer, ultimately reaching or surpassing human-level performance. In Soccer and Drawer Open, it learned slightly slower but still matched human performance. Notably, in Straighten Rope, CoRe achieved the fastest learning and highest stability, ultimately exceeding human performance. Overall, CoRe enables faster and more stable policy learning. As summarized in Table 1, CoRe achieves an average success rate of 99.0% in seven MetaWorld tasks, outperforming Sparse (69.6%), CLIP Score (4.8%), Eureka (75.5%), Text2Reward (74.2%), RL-VLM-F (56.8%), PrefVLM (18.6%) and ERL-VLM (37.0%), and approaching Human (99.3%). In SoftGym tasks, CoRe attains $-0.10$ in Fold Cloth, $-30.0$ in Pass Water, and 20.6 in Straighten Rope—the best among all methods. We observe that most methods achieve near-perfect performance on simple tasks, but struggle with complex ones. For example, Drawer Open rewards depend mainly on ball position and can be easily specified by Sparse, Human, and LLM/VLM-based methods, all of which achieve 100% except CLIP Score and Pre-

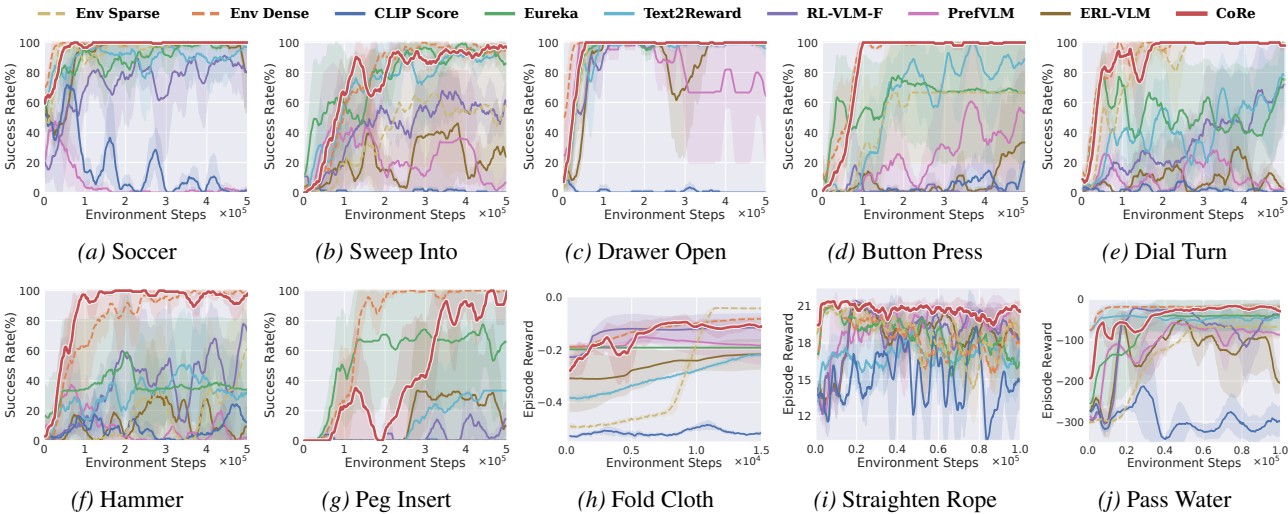

*Figure 3.* Learning curves on ten robotic manipulation tasks as measured by success rate and episode reward. The solid line and shaded regions represent the mean and standard deviation, respectively. Each task runs three times using different seeds.

*Table 1.* Comparison of final success rate and episode reward across ten tasks

| Method | Success Rate (%) | | | | | | | Episode Reward | | |
| --- | --- | --- | --- | --- | --- | --- | --- | --- | --- | --- |
| | Soccer | Sweep Into | Drawer Open | Button Press | Dial Turn | Hammer | Peg Insert | Fold Cloth | Straighten Rope | Pass Water |
| Env Sparse | 100.0 | 60.0 | 100.0 | 66.7 | 100.0 | 60.7 | 0.0 | **−0.04** | 18.6 | −67.9 |
| Env Dense | 100.0 | **98.0** | 100.0 | 100.0 | **100.0** | 97.3 | 100.0 | −0.08 | 18.1 | **−18.3** |
| CLIP Score | 1.3 | 0.0 | 0.0 | 20.7 | 0.0 | 11.3 | 0.0 | −0.52 | 15.0 | −299.4 |
| Eureka | 100.0 | 86.0 | 100.0 | 66.7 | 76.0 | 33.3 | 66.7 | −0.19 | 16.2 | −38.8 |
| Text2Reward | 96.7 | 96.0 | 96.0 | 88.0 | 78.0 | 32.0 | 33.3 | −0.22 | 17.4 | −43.7 |
| RL-VLM-F | 80.0 | 58.0 | 100.0 | 0.0 | 72.0 | 72.7 | 14.7 | −0.12 | 17.9 | −36.4 |
| PrefVLM | 1.3 | 6.7 | 64.0 | 54.0 | 2.7 | 1.3 | 0.0 | −0.18 | 20.5 | −86.9 |
| ERL-VLM | 80.7 | 24.0 | 100.0 | 33.3 | 2.7 | 9.3 | 9.3 | −0.22 | 18.1 | −202.4 |
| **CoRe** | **100.0** | 97.3 | **100.0** | **100.0** | 98.0 | **98.0** | **100.0** | −0.10 | **20.6** | −30.0 |

fVLM, which suffer from the data shift in pre-trained VLMs. In contrast, tasks like Hammer and Peg Insert involve intricate actions (grasping, positioning, aligning), where Sparse rewards are ineffective. Hammer's 66.7% reflects reward hacking (striking with the handle), while Peg Insert remains at 0%. Eureka and Text2Reward struggle to design suitable reward components and weights (33.3% and 32.0% in Hammer, respectively). Although Eureka learns fast in the early training because of its structured reward design from LLM in Peg Insert (rapidly achieve a 70% success rate), but its performance could not be further improved when not finding the implicit rewards. RL-VLM-F suffers from limited feedback and unstable reward learning (14.7% in Peg Insert). The rating used by ERL-VLM struggles to accurately assign credit for these tasks (9.3% in Hammer and Peg Insert). The learning rate of CoRe may not be as fast as structured-rewards-design method (as in Peg Insert), but accurate rewards estimation from FRM and RRM still make CoRe achieved the highest success rate (nearly 100

%). Collectively, most baselines exhibit large variance due to incomplete reward specification or reward instability. By contrast, CoRe excels in simple and complex tasks, owing to the complementary synergy of its two modules.

As shown in Table 3, CoRe demonstrates superior sample efficiency. Across tasks, it uses 0.15K–0.5K labels, compared to 0.5K–21K for other methods, corresponding to a 3-40× reduction. For example, in Fold Cloth, RL-VLM-F uses 500 labels while CoRe uses 150 ($\approx 3\times$); in MetaWorld, PrefVLM uses 21K labels while CoRe uses 0.5K ($\approx 40\times$). Other methods require more labels due to limited information in image preferences, noisy labels or coarse ratings which can hinder fine-grained reward learning.

As shown in Table 4, a full CoRe training run requires approximately 2.00M tokens, $0.37 API cost, and 2.15 hours of wall-clock time on average across the above ten tasks. Compared with methods such as RL-VLM-F and ERL-VLM, CoRe achieves a more favorable trade-off between

*Table 2.* Comparison of final success rate and episode reward across ten tasks in ablation experiments

| Method | Success Rate (%) | | | | | | | Episode Reward | | |
| --- | --- | --- | --- | --- | --- | --- | --- | --- | --- | --- |
| | Soccer | Sweep Into | Drawer Open | Button Press | Dial Turn | Hammer | Peg Insert | Fold Cloth | Straighten Rope | Pass Water |
| FRM$^0$ | 100.0 | 36.0 | 100.0 | 17.3 | 34.7 | 34.4 | 50.0 | $-0.68$ | 15.8 | $-88.1$ |
| FRM$^2$ | 100.0 | 93.3 | 100.0 | 34.7 | 79.3 | 37.3 | 65.3 | $-0.49$ | 17.0 | $-60.7$ |
| FRM$^4$ | 100.0 | 93.3 | 100.0 | 59.3 | 89.0 | 54.0 | 88.0 | $-0.32$ | 18.6 | $-35.8$ |
| RRM w/o imp | 100.0 | 75.3 | 71.3 | 3.3 | 33.5 | 12.7 | 0.0 | $-0.15$ | 17.4 | $-53.3$ |
| RRM | 100.0 | 94.0 | 88.7 | 8.7 | 36.0 | 49.3 | 36.0 | $-0.14$ | 19.0 | $-33.2$ |
| **CoRe** | **100.0** | **97.3** | **100.0** | **100.0** | **98.0** | **98.0** | **100.0** | $\mathbf{-0.10}$ | **20.6** | $\mathbf{-30.0}$ |

*Table 3.* Feedback numbers across ten tasks

| Method | MetaWorld | | Fold Cloth | | Straighten Rope / Pass Water | |
| --- | --- | --- | --- | --- | --- | --- |
| | Label | Image | Label | Image | Label | Image |
| RL-VLM-F | 5.0K | 10.0K | 0.50K | 1.00K | 2.0K | 4.0K |
| PrefVLM | 21.0K | 2.0M | 1.79K | 10.74K | 4.1K | 0.4M |
| ERL-VLM | 4.9K | 4.9K | 0.75K | **0.75K** | 0.9K | **0.9K** |
| **CoRe** | **0.5K** | **4.0K** | **0.15K** | 1.20K | **0.2K** | 1.6K |

*Table 4.* Average training cost across ten tasks

| Method | Token (M) | API Cost ($) | Time (h) |
| --- | --- | --- | --- |
| SAC | — | — | 0.97 |
| CLIP Score | — | — | 2.58 |
| Eureka | 0.03 | 0.03 | 1.87 |
| Text2Reward | 0.03 | 0.02 | 5.73 |
| RL-VLM-F | 5.50 | 0.79 | 6.72 |
| PrefVLM | — | — | 1.82 |
| ERL-VLM | 3.19 | 0.55 | 5.80 |
| CoRe | 2.00 | 0.37 | 2.15 |

performance, computational cost, and runtime efficiency. This efficiency mainly stems from the structured reward with Reward-Preference Alignment in FRM and video-level residual reward learning in RRM.

### 4.3. Ablation Study

**Effects of FRM and RRM.** To disentangle the contributions of CoRe's two components, we conducted an ablation study across 10 tasks. Table 2 shows that FRM (FRM$^4$) generally performs better in MetaWorld tasks, while RRM performs better in Softgym tasks. This suggests that code-based FRM is effective for rigid-object manipulation but limited in expressiveness, making it insufficient for deformable objects. Conversely, RRM leverages vision-based reward models to handle deformable tasks but struggles with complex long-horizon ones due to limited feedback. By combining them, CoRe achieves consistent gains: RRM complements FRM's

limited expressiveness, while FRM facilitates RRM's reward learning. The two modules are thus mutually reinforcing, enabling more efficient, robust policy learning.

**Reward-Preference Alignment in FRM.** We further ablate the number of Reward-Preference Alignment iterations in FRM. FRM$^i$ denotes the $i$-th alignment, and policy success rates are reported in Table 2. We find that alignment consistently improves policy. For example, in Sweep Into and Dial Turn, FRM$^0$ achieved only 30% success, but after one alignment, success rates rose to 93.3% and 79.3%, respectively. However, FRM is sensitive to task difficulty and label noise: in Hammer and Fold Cloth, the improvement remains limited. Overall, Reward-Preference Alignment enables fast validation of reward candidates, ensures consistency with human preference, and drives continuous improvement.

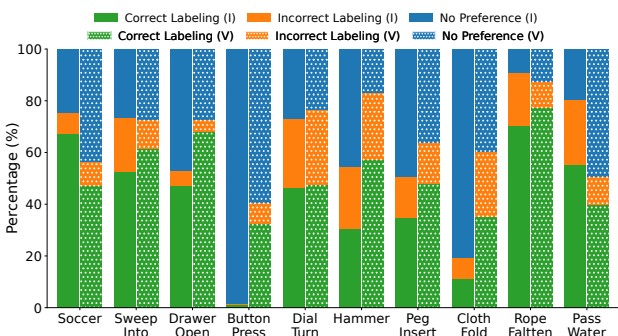

*Figure 4.* Measure the accuracy of VLMs in terms of image-level preference (**I**) and video-level preference (**V**) with task completion. Correct Labeling indicates that VLMs favor the image or video with higher task completion. Incorrect Labeling denotes that VLMs fail to identify the option with higher task completion. No Preference indicates that VLMs cannot provide a preference.

**VLMs preference for images and videos.** We evaluate the labeling accuracy of VLMs under three conditions: Correct, Incorrect, and No Preference. As shown in Figure 4, we find that video preference consistently outperforms image preference. The average results of video-level preference labels across all tasks achieve 51.5% Correct, 15.0% Incorrect, and 33.5% No Preference, while image-level preference labels reach only 41.6%, 15.6%, and 42.8%, respectively. This im-

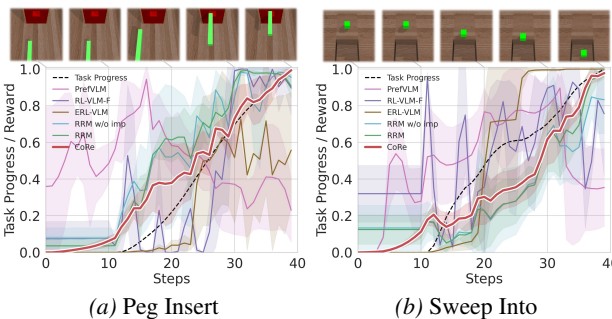

*Figure 5.* The alignment of learned rewards and task progress on the expert trajectory. The learned rewards are averaged over three trained reward models with different seeds, and the shaded region represents the standard deviation.

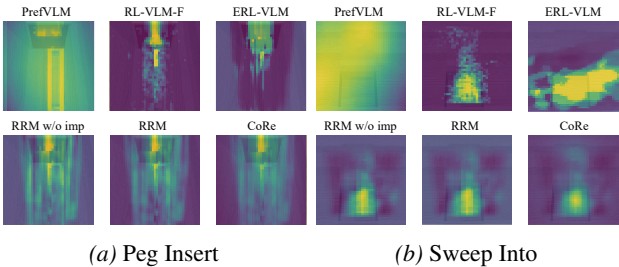

*Figure 6.* We visualize the learned rewards corresponding to objects at different positions as a 2D heat map. Brighter indicates a higher reward. The rewards are averaged over three trained reward models with different seeds.

*Table 5.* Simulation and real-world results on five tasks.

| Method | Drawer Open ↑ | | Dial Turn ↑ | | Hammer ↑ | | Fold Cloth ↓ | | Straighten Rope ↑ | |
|---|---|---|---|---|---|---|---|---|---|---|
| | sim | real | sim | real | sim | real | sim | real | sim | real |
| CLIP Score | 0 | 0 | 0 | 0 | 0 | 0 | 11.0 | 15.6 | 55 | 33 |
| Eureka | 100 | 70 | 60 | 45 | 0 | 0 | 4.8 | 7.3 | 76 | 69 |
| Text2Rward | 100 | 75 | 55 | 45 | 40 | 25 | 5.1 | 7.0 | 51 | 42 |
| PrefVLM | 85 | 55 | 0 | 0 | 5 | 0 | 2.9 | 4.5 | 81 | 71 |
| RL-VLM-F | 100 | 65 | 55 | 45 | 50 | 35 | 2.5 | 4.9 | 83 | 73 |
| ERL-VLM | 100 | 80 | 0 | 0 | 0 | 0 | 4.2 | 5.2 | 81 | 53 |
| **CoRe** | **100** | **90** | **75** | **70** | **90** | **80** | **2.1** | **3.2** | **90** | **84** |

provement stems from our video-level preference prompts, which compare every frame and better capture task-relevant details, yielding higher-quality preference labels.

To evaluate whether VLM-derived importance weights reflect true state importance, we measure their Spearman correlation with task progress (Spearman, 1961). Across 10 tasks, the average $\rho$ is 0.48, indicating moderate alignment. However, directly using these weights as rewards leads to high variance due to noise (Wang et al., 2024). Instead, RRM treats preference prediction as the primary objective and uses importance weights only as an auxiliary constraint, improving reward accuracy, efficiency, and stability.

**Alignment of rewards with task progress.** We further assess the quality of the learned rewards from both temporal and spatial perspectives. Specifically, we examine their alignment with task progress along expert trajectories and distribution across object positions. We compare Pre-fVLM, RL-VLM-F, ERL-VLM, RRM without importance distribution (RRM w/o imp), RRM, and CoRe. Except for PrefVLM, all others are image-based rewards. For each method, normalized rewards (0–1) are computed and visualized for Peg Insert and Sweep Into in Figure 5 and Figure 6. PrefVLM, RL-VLM-F, and ERL-VLM suffer from inaccurate rewards and significant fluctuations, with spatial distri-

butions that remain unsmooth and imprecise. RRM w/o imp leverages video-level preference to achieve more accurate rewards and yields smoother reward distributions; however, credit assignment under limited feedback still introduces some noise. RRM further improves accuracy and stability, generating rewards with near-linear correlation with task progress and consistent spatial smoothness. With the integration of FRM, CoRe yields the most accurate and stable rewards, showing strong consistency with task progress and the smoothest spatial distribution, thereby facilitating stable policy optimization. Quantitatively, Spearman correlations between learned rewards and task progress along expert trajectories averaged over ten tasks are 0.09 (PrefVLM), 0.44 (RL-VLM-F), 0.48 (ERL-VLM), 0.56 (RRM w/o imp), 0.77 (RRM), and 0.88 (CoRe), confirming that CoRe provides the most accurate and reliable rewards. These results highlight that incorporating importance weights and the FRM yields stable, precise reward models, with improvements also reflected in policy performance (Table 2).

### 4.4. Deploy on the Real Robot

We deploy policies trained via CLIP Score, Eureka, Text2Reward, PrefVLM, RL-VLM-F, ERL-VLM and CoRe on UR5 robotic arms across five real-world tasks: Drawer Open, Dial Turn, Hammer, Fold Cloth, and Straighten Rope (Figure 8). For Drawer Open, Dial Turn and Hammer, we reconfigure simulation environments to match real setups and use Aruco markers for object pose estimation. Success rate (higher is better) is used as the performance metric. For Fold Cloth and Straighten Rope, key points are tracked via color detection (e.g., cloth corners in Fold Cloth, rope endpoints and midpoint in Straighten Rope). Task-specific geometric metrics are used for evaluation: in Straighten Rope, the task completion percentage (higher is better) is defined as the rope length at the end of an episode relative to its maximum length; in Fold Cloth, the performance metric is measured as the positional error (cm, lower is better) of the three corner points corresponding to the diagonal cloth at the end of an episode. To validate the policy's robustness when

transferring from sim to real, they were directly executed on the UR5 without fine-tuning. All methods were repeated twenty times. As summarized in Table 5, CoRe consistently outperforms other methods across all tasks, demonstrating not only superior performance in simulation but also robustness to real-world noise with high-quality reward signals, thereby enabling reliable robotic deployment.

## 5. Conclusion and Future Work

In this paper, we introduced CoRe, a framework that decomposes reinforcement learning rewards into formal rewards and residual rewards to achieve preference-aligned policy learning in robotic manipulation. The Formal Reward Module (FRM) leverages LLMs and preference to iteratively design and refine code-based formal rewards, while the Residual Reward Module (RRM) exploits video-level preference and importance weight from VLMs to capture fine-grained alignment with human intentions. Through their complementary synergy, CoRe enables efficient and robust rewards construction without manual reward engineering. Extensive experiments in both simulation and the real world demonstrate that CoRe achieves superior performance, stability, and human alignment compared with existing methods.

Due to the reliance on iterative LLM/VLMs querying, future work will explore uncertainty-aware reward modeling and more efficient querying strategies to further improve the robustness and scalability of reward learning. In addition, evaluating CoRe in more diverse and unstructured real-world environments is an important direction for future research, which may further improve its generalizability across complex manipulation scenarios.

## Acknowledgements

This work was supported in part by New Generation Artificial Intelligence-National Science and Technology Major Project 2025ZD0122904, in part by National Natural Science Foundation of China(Grant Number:62473366) and in part by National Science and Technology Major Project 2026ZD1609003.

## Impact Statement

This work presents CoRe, a reward learning framework for robotic manipulation that reduces the need for manually designed reward functions in RL and improves the alignment of policies with human preferences. As CoRe relies on LLMs/VLMs for reward generation and preference feedback, potential biases or inaccuracies in their outputs may propagate into the learned reward functions and policies. In addition, iterative querying of LLMs/VLMs introduces additional computational and financial costs during training.

These factors may raise robustness, scalability, and safety considerations in real-world robotic applications.

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

# Appendix

## A. Details on Tasks and Environments

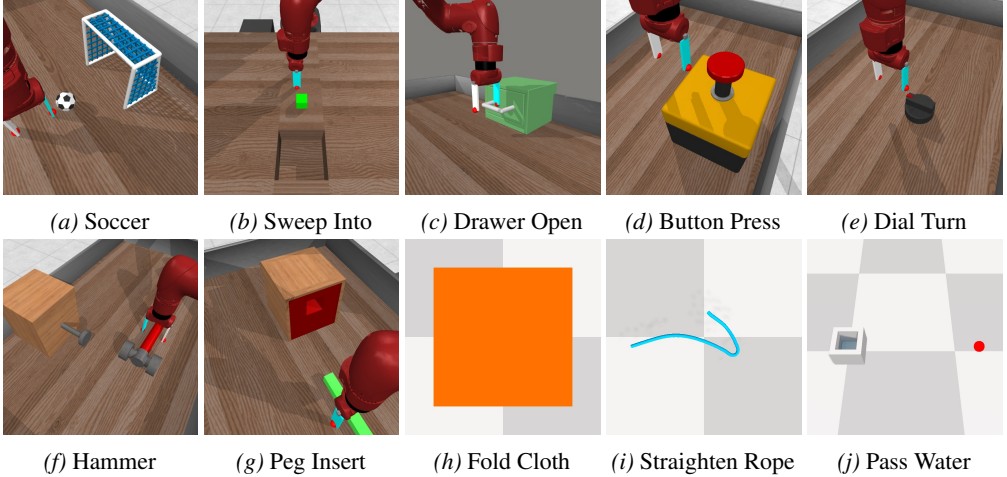

*(a)* Soccer    *(b)* Sweep Into    *(c)* Drawer Open    *(d)* Button Press    *(e)* Dial Turn

*(f)* Hammer    *(g)* Peg Insert    *(h)* Fold Cloth    *(i)* Straighten Rope    *(j)* Pass Water

*Figure 7.* Ten simulation tasks. The first seven images depict rigid and articulated object manipulation tasks from MetaWorld. The next three images illustrate deformable object manipulation tasks from SoftGym.

We evaluate our method on ten simulated and five real-world robotic manipulation tasks involving rigid, articulated, and deformable objects as shown in Figure 7 and Figure 8, which together cover a wide range of primitive actions such as pushing, pulling, pressing, twisting, and grabbing.

### A.1. MetaWorld Tasks

Seven tasks from MetaWorld (Yu et al., 2020) involving the manipulation of rigid and articulated objects on the table, with a simulated Sawyer robot.

- *Soccer*: The goal is to kick the ball into the goal.

- *Sweep Into*: The goal is to sweep a green puck into a hole.

- *Drawer Open*: The goal is to pull the green drawer completely out.

- *Button Press*: The goal is to press the red button from top to bottom. The robot needs to keep pressing down because there's a spring that will automatically push the button back up.

- *Dial Turn*: The goal is to rotate a dial 180 degrees counterclockwise.

- *Hammer*: The goal is to grasp a hammer and hammer a gray nail. The robot needs to hit the nail with the hammerhead, not the handle.

- *Peg Insert*: The goal is to grasp a green peg and insert it horizontally into the hole in the block.

**Observation Space.** We use the SAC algorithm and state-based observations for policy learning, and high-dimensional RGB images rendered by a simulator for image-based reward learning. Following the original MetaWorld settings (Yu et al., 2020), the state observation is a 39-dimensional vector, consisting of the 3D position of the end-effector, gripper status, the 3D position and quaternion of target objects, and the 3D position of the goal. Following RL-VLM-F (Wang et al., 2024), we set the resolution of the image observation to 300×300. We also render only task-relevant objects by setting the robot arm transparent to mitigate VLM hallucinations and promote visual understanding. The camera view is adjusted to focus on objects, and the initial state is modified to bring the end-effector closer to them. The example of image observation is shown in Figure 10.

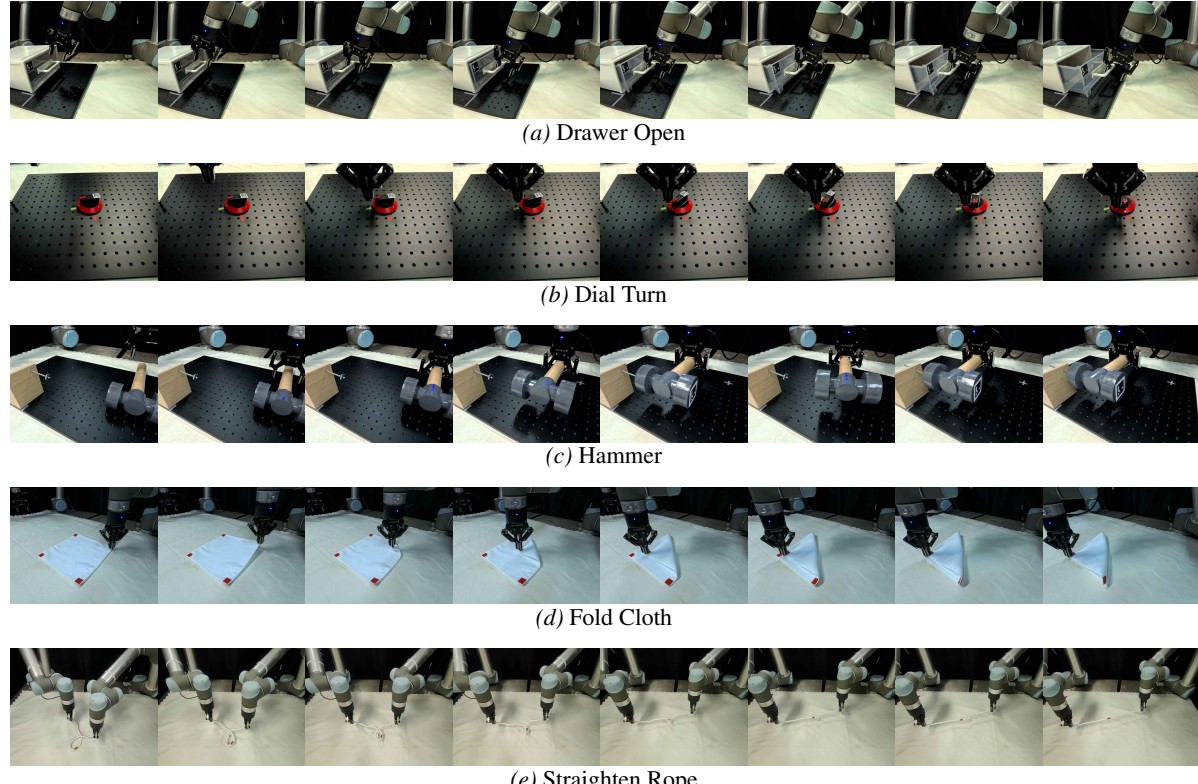

*(a)* Drawer Open

*(b)* Dial Turn

*(c)* Hammer

*(d)* Fold Cloth

*(e)* Straighten Rope

*Figure 8.* Five real-world tasks. Execution sequence of policy on the real-world UR5 robotic arm.

**Action Space.** The action in MetaWorld tasks is a 4-dimensional vector consisting of the 3D position displacement of the end-effector and the torque the gripper fingers should apply. All dimensions of the action are normalized to [-1, 1].

### A.2. SoftGym Tasks

Three tasks from SoftGym (Lin et al., 2021) involving the manipulation of deformable objects on a table following (Wang et al., 2024).

- *Fold Cloth*: The goal is to fold a cloth diagonally from the upper left corner to the lower right corner. There is a transparent picker that is always fixed to the upper left corner of the cloth.

- *Straighten Rope*: The goal is to straighten a rope from a random configuration. The rope is controlled by two transparent pickers, which are fixed to its endpoints at the start of the task.

- *Pass Water*: The goal is to pass a glass of water to the designated location (the red dot) while ensuring that no water spills out. The glass can only move in a fixed direction.

**Observation Space.**

The settings for policy and reward learning are the same as in MetaWorld, but the state observation and the image observation resolutions are set differently. We all follow the RL-VLM-F settings.

- *Fold Cloth*: The state observation consists of the positions of a subset of particles from the cloth mesh. The cloth has an original resolution of $40{\times}40$ particles, which is uniformly subsampled to a $6 \times 6$ grid. The final state includes the 3D position of the picker (3) and the 3D positions of all subsampled cloth particles (108), resulting in a 111-dimensional vector. The resolution of the image observation is $360{\times}360$.

- *Straighten Rope*: The state observation comprises the positions of all particles along the rope (10 particles, 30) and the two pickers (6), resulting in a 36-dimensional vector. The resolution of the image observation is $240{\times}240$.

| Task Name | Task Description |
|---|---|
| *Soccer* | to move the soccer ball into the goal. |
| *Sweep Into* | to minimize the distance between the green cube and the hole. |
| *Drawer Open* | to open the drawer. |
| *Button Press* | to press the red button down completely from top to bottom |
| *Dial Turn* | to turn the red line to the bottom of the dial |
| *Hammer* | to hammer the grey nail completely in with a red hammer |
| *Peg Insert* | to insert the green peg into the hole of the red block |
| *Fold Cloth* | to fold the cloth diagonally from the top left corner to the bottom right corner |
| *Straighten Rope* | to straighten the blue rope |
| *Pass Water* | to move the container, which holds water, to be as close to the red circle as possible without causing too many water droplets to spill |

*Table 6.* Task descriptions of CoRe.

- *Pass Water*: The state observation includes the container dimensions (width, length, and height), the target container position, the water height within the container, and the amounts of water inside and outside the container, yielding a 7-dimensional state space. The resolution of the image observation is 360×360.

**Action Space.**

All dimensions of the action are also normalized to [-1, 1]

- *Fold Cloth*: We employ a pick-and-place action primitive. The cloth is assumed to be grasped at the top left corner at initialization, and the action specifies a 2D target placement location, resulting in a 2-dimensional vector.

- *Straighten Rope*: Two pickers are used to manipulate the rope, one attached to each end. Accordingly, the action space consists of the 3D displacement vectors for both pickers, resulting in a 6-dimensional action space. The rope endpoints are assumed to be grasped at the start of each episode.

- *Pass Water*: The motion of the container is constrained to a single dimension; thus, the action is a 1-dimensional scalar representing the displacement of the container along that axis.

**A.3. Real Robot Taks**

To validate the policies robustness when transferring from sim to real, we designed five real-world operational tasks based on the simulation task, as shown in Figure 8. We first simulated real-world scenarios in a simulation environment, then retrained the policy and directly deployed it to the UR5 robot to perform the manipulation tasks.

**Observation Space.** For Drawer Open, Dial Turn and Hammer, the state observation of policy inputs is consistent with the settings in the simulation, where the 3D position of the end-effector and the gripper status are obtained in real-time through the Python SDK of UR5. The 3D position and quaternion of the object are obtained using visual detection of Aruco markers. We use the Intel RealSense D435i depth camera, intrinsic and extrinsic parameter calibration for accurate pose estimation. For Fold Cloth, the state observation is a 15-dimensional vector, consisting of the 3D position of the four corner points of the cloth and the end-effector. For Rope Straighten, the state observation is a 15-dimensional vector, consisting of the 3D position of two endpoints and midpoints of the rope followed by the 3D position of two end-effectors. Since we marked the keypoints of the cloth and rope, it was easy to obtain the 3D positions of the keypoints using visual color detection and the depth camera.

**Action Space.**

The action settings are basically consistent with the simulation. However, due to the limitations of the Robotiq 2F-85 gripper, we used the absolute position of the gripper instead of torque in the simulation.

---

**Algorithm 1** CoRe

---

**Input:** Task description $t_d$, definition of state and action $t_s$, task start image $I_s$ and task goal image $I_g$.

1: Initialize policy $\pi_\theta$ and reward $\hat{r}_\psi$, replay buffer $\mathcal{B} \leftarrow \emptyset$, preference buffer $\mathcal{D} \leftarrow \emptyset$, formal reward update frequency $M$, VLM query frequency $N$, number of queries $H$ per feedback session.
2: **for** each iteration **do**
3:    // POLICY LEARNING AND DATA COLLECTION
4:    Obtain state $s_{t+1}$, image $I_{t+1}$ by taking $a_t \sim \pi_\phi(a_t|s_t)$
5:    Store transition $\mathcal{B} \leftarrow \mathcal{B} \cup \left\{(s_t, I_t, a_t, s_{t+1}, I_{t+1}, \hat{r}_{best}^{n-1}(t)) + \hat{r}_\psi(t)\right\}$
6:    **for** each gradient step **do**
7:      Sample random batch $\left\{(s_t, a_t, s_{t+1}, \hat{r}_{best}^{n-1}(t)) + \hat{r}_\psi(t)\right\}_{j=1}^{B} \sim \mathcal{B}$
8:      Update policy $\pi_\theta$ with any off-policy RL algorithm
9:    **end for**
10:   // FORMAL REWARD UPDATING
11:   **if** iteration % $M$ == 0 **then**
12:     Sample reward candidates $\hat{r}^n$ based on $t_d$, $t_s$, $\hat{r}_{best}^{n-1}$ and $t_f^{n-1}$
13:     Select formal reward according to Equation (2)
14:     Relabel entire replay buffer $\mathcal{B}$
15:   **end if**
16:   // PREFERENCE BY VLM AND REWARD LEARNING
17:   **if** iteration % $N$ == 0 **then**
18:     **for** $h = 1$ to $H$ **do**
19:       Randomly sample two segments $(\sigma^0, \sigma^1)$ from $\mathcal{B}$
20:       Query VLM for label $y$ and obtain importance weight $(W^0, W^1)$ according to Equation (4)
21:       Store preference $\mathcal{D} \leftarrow \{(\sigma^0, \sigma^1, y, W^0, W^1)\}$
22:     **end for**
23:     **for** each gradient step **do**
24:       Sample minibatch $\{(\sigma^0, \sigma^1, y, W^0, W^1)_j\}_{j=1}^{\mathcal{D}} \sim \mathcal{D}$
25:       Update $\hat{r}_\psi$ in Equation (5) with respect to $\psi$
26:     **end for**
27:     Relabel entire replay buffer $\mathcal{B}$
28:   **end if**
29: **end for**

---

# B. Training Details and Hyperparameters

## B.1. Reward Optimization

### Formal Reward Optimization.

We followed the iterative process of Eureka's reward design (Ma et al., 2024), except for the reward search and reward design prompts. For reward search, Eureka uses task completion to select the best reward candidates defined in Equation (1), while our method uses Reward-Preference Alignment defined in Equation (2). For reward design prompts, Eureka needs access to all the Python code of the environment to obtain the complex reward function. Because of the learnable residual rewards, our method does not require the full Python code; it only needs some descriptions of the reward function, such as the definitions of the input and output variables of the reward function (state, action and target), which are easy to design, as shown in Table 15. Our method performs five reward searches, sampling four reward candidates each time by using GPT-4.1 mini (OpenAI, 2025).

### Image-based Residual Reward Learning.

For the image-based reward model, we employ a 4-layer convolutional neural network for MetaWorld tasks and a standard ResNet-18 (He et al., 2016) for SoftGym tasks following (Wang et al., 2024). Following PEBBLE (Lee et al., 2021a), we use an ensemble of three reward models, with a tanh activation applied to the output reward. All models are optimized using Adam (Kingma & Ba, 2014) with an initial learning rate of $3 \times 10^{-4}$. The ResNet-based reward model has a batch size of 8, while others have a batch size of 32. The query selection scheme is uniform sampling. The loss for reward model optimization consists of the cross-entropy loss of preference labels and the KL divergence loss of importance weights defined in Equation (5). For MetaWorld tasks, we use a segment size of 20 and downsample to 3 for VLM querying (Gemini 2.0 Flash (Comanici et al., 2025)).

*Table 7.* Hyperparameters for reward learning.

| Parameter | Meta-World | SoftGym |
|---|---|---|
| Train Steps | 500000 | 15000 (Cloth) and 100000 (Rope and Water) |
| Initialize Formal Reward Steps | 19000 | 250 (Cloth) and 9000 (Rope and Water) |
| Random Steps | 1000 | 250 (Cloth) and 1000 (Rope and Water) |
| Formal Reward Update Frequency $M$ | 25000 | 2500 (Cloth) and 25000 (Rope and Water) |
| VLM Query Frequency $N$ | 4000 | 1000 (Cloth) and 5000 (Rope and Water) |
| Numer of Queries $H$ Per Session | 20 | 10 |
| Segment | 20 | 3 |
| Max Feedback | 500 | 150 (Cloth) and 200 (Rope and Water) |
| KL Weight | 5 | 5 |
| Batch Size | 32 | 8 |

## B.2. Policy Learning

Following PEBBLE (Lee et al., 2021a) and RL-VLM-F (Wang et al., 2024), we adopt Soft Actor-Critic (SAC) (Haarnoja et al., 2018) as the off-policy learning algorithm. We use the same actor–critic network architectures and hyperparameter settings as in the original work. The policy is learned with state observation for all methods. Due to the formalized reward function, our method directly optimizes the policy without requiring unsupervised pre-training.

## B.3. Training Details

The full procedure of our method is summarized in Algorithm 1. Policy learning and reward learning are performed simultaneously: the policy is optimized using experiences from the replay buffer and the learned reward model, while the reward model is updated based on policy performance and preference labels over trajectory segments provided by VLMs. The optimization of both the formal reward and the residual reward is also carried out in parallel. Specifically, prior to training, we perform a task-completion-based reward search to obtain an initial formal reward. During training, the policy collects experience and updates online at each time step. The formal reward is updated every $M$ time steps, while VLM-based preference labeling and residual reward updates are conducted every $N$ time steps. After each reward update, all experiences stored in the replay buffer are relabeled accordingly.

*Table 8.* Hyperparameters for policy learning.

| Parameter | Fold Cloth | Others |
|---|---|---|
| Number of layers | 3 | 2 |
| Hidden units per each layer | 1024 | 256 |
| Learning Rate | $5 \times 10^{-4}$ | $3 \times 10^{-4}$ |
| Batch Size | 1024 | 512 |
| Init Temperature | 0.1 | 0.1 |
| Critic Target Update Frequency | 2 | 2 |
| Optimizer | Adam | Adam |
| $(\beta^1, \beta^2)$ | (.9, .999) | (.9, .999) |
| Discount | 0.99 | 0.99 |
| Critic EMA $\tau$ | 0.005 | 0.005 |

## C. Baselines

To verify the performance, we compare CoRe with existing methods that also leverage pretrained LLM/VLM to directly generate reward score, reward code or preference feedback. The reward code method requires the environment code and the task description. The reward score and preference feedback require the task description and the agent's image observations. The reward score is calculated directly through similarity, while preference feedback involves learning the reward model from the feedback.

- **CLIP Score** (Rocamonde et al., 2024): The reward score is calculated as the cosine similarity between the embeddings of the task description and the agent's current image observation in the CLIP (Radford et al., 2021) latent space. We run this method through the implementation of RL-VLM-F [1].

- **Eureka** (Ma et al., 2024): This baseline is the LLM-driven code-based reward generation method. It needs to access the environment code, automatically design reward functions based on task description, and continuously optimize them through reward reflection. We extended this method to the MetaWorld and SoftGym tasks based on the original implementation of Eureka [2].

- **Text2Rward** (Xie et al., 2024): Similar to Eureka, LLMs generate reward functions based on the environment code and task description, but this baseline requires human natural language feedback to optimize the reward function. We performed five rounds of reward optimization based on the original implementation of Text2Reward [3] and our online human feedback. The final reward function was used to train the policy.

- **RL-VLM-F** (Wang et al., 2024): Similar to the PbRL framework, but the preference feedback is obtained by querying VLMs based on the task description and two agents' image observations. The image-based reward model is trained on these image-level preference labels. We will run this method directly through the original implementation of RL-VLM-F.

- **PrefVLM** (Ghosh et al., 2025): Instead of querying VLMs ((Ma et al., 2023)) for the image, this baseline first uses the similarity reward score of VLM to obtain video-level preference. The noise labels from VLMs are selected through a noise filtering (Cheng et al., 2024) and then provided to human annotators which then is used to fine-tune VLM. Since the original implementation is not publicly available, we reimplemented the method based on the paper and the PEBBLE codebase [4], keeping all architectural choices and hyperparameters consistent with those reported in the original work.

- **ERL-VLM** (Luu et al., 2025): Query VLM with an image or a sequence of images to obtain the rating based on task description (discrete, e.g., bad, average, good), and then use the ratings to train an image-based reward model for rating-based RL (White et al., 2024). We directly ran the method using the original implementation of ERL-VLM [5].

- **Env Sparse**: Environment Sparse Reward indicates that the reward for task success is 1, and all other rewards are 0.

- **Env Dense**: Environment Dense Reward is the dense reward provided by the benchmark.

To ensure fairness, all reward function generation methods (Eureka, Text2Reward and our method) are powered by GPT-4.1 mini (OpenAI, 2025), and all preference labels (RL-VLM-F, ERL-VLM and our method) are provided by Gemini 2.0 Flash (Comanici et al., 2025). The parameters related to reward construction are consistent with those in the original paper.

## D. Additional Experiments

### D.1. Analysis with Stronger Models

We further investigate whether simply scaling to stronger models can compensate for the limitations of single-form reward learning. As shown in Table 9, stronger models improve the performance of FRM and RRM to some extent, but the gains are relatively limited compared with the increased API cost(e.g., FRM: 54.0% → 54.7% with 9× cost; RRM: 49.3% → 62.6% with 6× cost). In contrast, CoRe achieves substantially better performance while maintaining favorable efficiency, suggesting that reward decomposition is more effective than solely scaling models.

---

[1] https://github.com/yufeiwang63/RL-VLM-F
[2] https://github.com/eureka-research/Eureka
[3] https://github.com/xlang-ai/text2reward
[4] https://github.com/rll-research/BPref
[5] https://github.com/tunglm2203/erlvlm

*Table 9.* Stronger models performance in Hammer

| Method | Success Rate(%) | Cost($) |
|---|---|---|
| FRM with GPT-4.1 mini | 54.0 | 0.06 |
| FRM with GPT-5.4 | 54.7 | 0.53 |
| RRM with Gemini 2.0 Flash | 49.3 | 0.31 |
| RRM with Gemini 3 Flash | 62.6 | 1.88 |
| CoRe with GPT-4.1 mini&Gemini 2.0 Flash | 97.3 | 0.37 |

## D.2. Experiments with Open-source VLMs

We additionally evaluate CoRe using open-source VLMs to verify that the framework is not restricted to proprietary models. As shown in Table 10, replacing Gemini 2.0 Flash with Qwen3-VL-8B-Instruct(Yang et al., 2025) still achieves strong performance across manipulation tasks (84.0% in Sweep Into, 88.7% in Hammer). In contrast, Video-LLaVA-7B (Lin et al., 2024a) fails to produce sufficiently reliable preference labels due to limited instruction-following and video understanding capabilities. These results suggest that CoRe is model-agnostic and can benefit from future advances in open-source models.

*Table 10.* Open-source VLMs performance

| Method | Sweep Into | Hammer |
|---|---|---|
| Video-LLaVa-7B | — | — |
| Qwen3-VL-8B-Instruct | 84.0% | 88.7% |
| Gemini 2.0 Flash | 97.3% | 98.0% |

## D.3. Robustness under Visual Disturbances

To further evaluate robustness under visual disturbances, we introduce a randomly moving red cube into the environment as shown in Figure 9. Table 11 shows that CoRe maintains stable performance with only minor degradation under noisy settings, demonstrating robustness to perceptual disturbances and environmental noise.

*Figure 9.* Visual disturbance scenarios

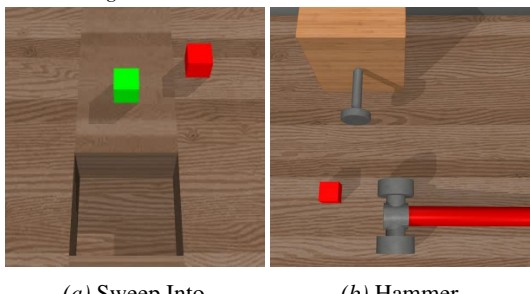

*(a)* Sweep Into      *(b)* Hammer

*Table 11.* Visual disturbances experiment performance

| Method | Sweep Into | Hammer |
|---|---|---|
| Clear | 97.3% | 98% |
| Noisy | 91.7% | 96% |

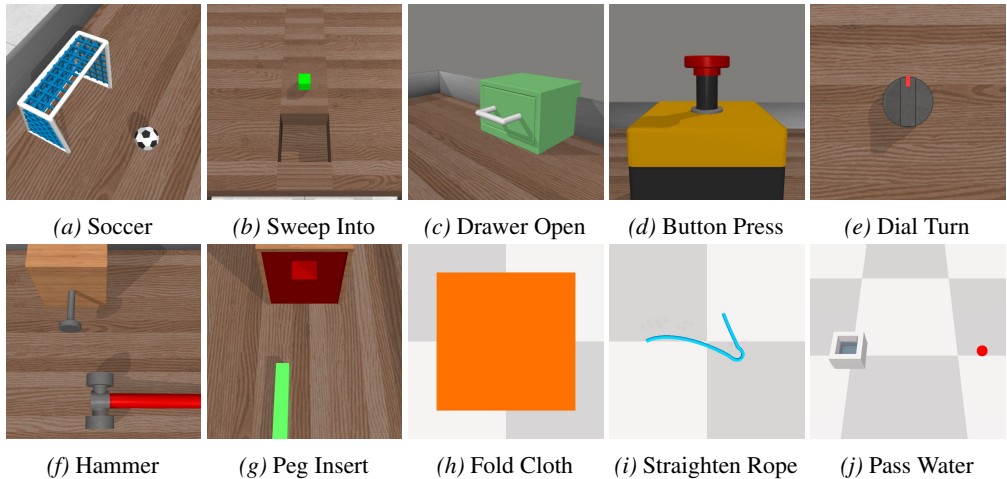

*(a)* Soccer    *(b)* Sweep Into    *(c)* Drawer Open    *(d)* Button Press    *(e)* Dial Turn

*(f)* Hammer    *(g)* Peg Insert    *(h)* Fold Cloth    *(i)* Straighten Rope    *(j)* Pass Water

*Figure 10.* The start images for the 10 simulation tasks.

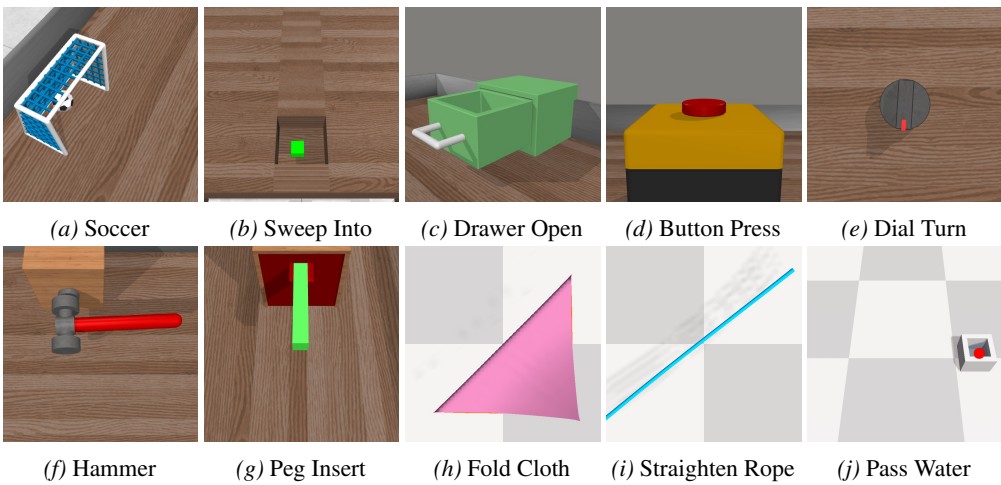

*(a)* Soccer    *(b)* Sweep Into    *(c)* Drawer Open    *(d)* Button Press    *(e)* Dial Turn

*(f)* Hammer    *(g)* Peg Insert    *(h)* Fold Cloth    *(i)* Straighten Rope    *(j)* Pass Water

*Figure 11.* The goal images for the 10 simulation tasks.

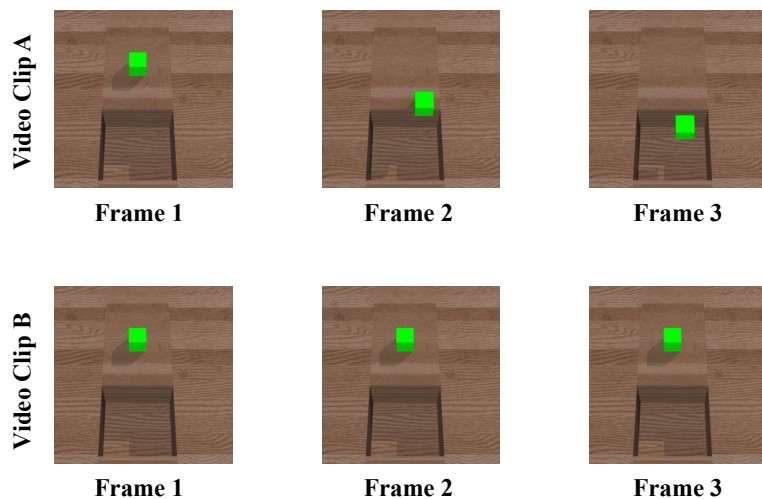

*Figure 12.* An example of a video-level preference label for Sweep Into.

VLM Analysis of a video-level preference label for Sweep Into as shown in Figure 12 fonttitle

**Analysis:**

Okay, let's analyze the video clips and compare them to the goal image.

**(1) What does each frame of video clip A show and what are the differences?**

* **Frame 1:** A green cube is positioned on a wooden surface, slightly above and to the left of a rectangular hole. A shadow is cast to the left of the cube. * **Frame 2:** The green cube has moved closer to the hole, positioned at the edge of the hole. * **Frame 3:** The green cube is now inside the hole.

The difference between the frames is the position of the green cube. It moves closer to and then into the hole in subsequent frames.

**(2) What does each frame of video clip B show and what are the differences?**

* **Frame 1:** A green cube is positioned on a wooden surface, slightly above and to the left of a rectangular hole. A shadow is cast to the left of the cube. * **Frame 2:** A green cube is positioned on a wooden surface, slightly above and to the left of a rectangular hole. A shadow is cast to the left of the cube. * **Frame 3:** A green cube is positioned on a wooden surface, slightly above and to the left of a rectangular hole. A shadow is cast to the left of the cube.

There is no difference between the frames. The green cube remains in the same position.

**(3) Is there any difference in each frame of the video clip and the goal?**

* **Video Clip A:** * **Frame 1:** The green cube is outside the hole, unlike the goal image where it's inside. * **Frame 2:** The green cube is partially inside the hole, unlike the goal image where it's fully inside. * **Frame 3:** The green cube is inside the hole, matching the goal image. * **Video Clip B:** * **Frame 1:** The green cube is outside the hole, unlike the goal image where it's inside. * **Frame 2:** The green cube is outside the hole, unlike the goal image where it's inside. * **Frame 3:** The green cube is outside the hole, unlike the goal image where it's inside.

In summary, Video clip A achieves the goal in frame 3, while video clip B does not achieve the goal in any of its frames.

**Preference:**

**decision#: **A

**evaluate goal completion for each frame in the video clip: ** A:[0.3, 0.6, 1] B:[0.3, 0.3, 0.3]

**why: ** Video A shows the cube moving towards and eventually inside the hole, thus minimizing the distance. Video B shows the cube remaining stationary and not minimizing the distance.

```python
import numpy as np
def reward_function(observation, action, target_pos):
    # Unpack observation components
    # observation = [x_hand, y_hand, z_hand, gripper_open, x_obj, y_obj, z_obj,
    # ↪  qx_obj, qy_obj, qz_obj, qw_obj]
    hand_pos = observation[0:3]
    # gripper_open = observation[3]  # Normalized gripper opening, not needed for
    # ↪  sweep task specifically
    obj_pos = observation[4:7]
    # obj_quat = observation[7:11]   # Not used in reward for sweeping

    # Distance from object to target (hole)
    dist_obj_to_target = np.linalg.norm(obj_pos - target_pos)

    # Distance from hand to object
    dist_hand_to_obj = np.linalg.norm(hand_pos - obj_pos)

    # We want to reward the cube being closer to the hole,
    # and encourage the hand to stay near the cube to sweep it.
    # Also encourage progress towards the goal.

    # Temperature parameters for exponential scaling
    temp_obj_to_target = 0.5
    temp_hand_to_obj = 1.0

    # Reward component: cube closer to target gives higher reward (range 0 to 1)
    # Apply negative distance scaled and exponentiated
    r_obj_to_target = np.exp(-dist_obj_to_target / temp_obj_to_target) - 0.5  # shift
    # ↪  to roughly center near 0

    # Reward component: hand close to object (to encourage interaction)
    r_hand_to_obj = np.exp(-dist_hand_to_obj / temp_hand_to_obj) - 0.5  # shift to
    # ↪  roughly center near 0

    # Combine rewards with weights
    # Since these scores range roughly in [-0.5, 0.5], sum ranges [-1, 1]
    reward = r_obj_to_target + r_hand_to_obj

    # Clip total reward to [-1, 1]
    reward = np.clip(reward, -1.0, 1.0)

    # Compose individual reward dict for info/debugging
    reward_components = {
        "cube_to_target": r_obj_to_target,
        "hand_to_cube": r_hand_to_obj,
    }

    return reward, reward_components
```

*Table 12.* An example of a formal reward function for Sweep Into.

---

**Prompt Template for Preference Label of CoRe**

---

**Analysis Template:** You will be given a task, and images showing the task at 0% (initial) and 100% (completed). The task is
`<Task Description>`.
The task initialization image is as follows: `<Start Image>`
The task completion image is as follows: `<Goal Image>`
You will now analyze two video clips (A and B) consisting of multiple frames each. Important evaluation rules:
1. A frame that moves away from the initial state but toward the goal state should be considered progress.
2. Do NOT treat frames that differ from the initialization image as worse by default. Always evaluate based on proximity to the task completion image.
3. The task progress score must be between 0.0 (same as initialization) and 1.0 (same as completion). Use intermediate scores like 0.3, 0.7, etc. Negative progress is not allowed.
4. Only describe objects that are directly involved in the task. Ignore irrelevant background elements or motion that is unrelated to task completion.
`<Video clip A>`
`<Video clip B>`
Now answer the following:
(1) For Video clip A: What are the differences between each frame of video clip A and the task initialization image and the task completion image?
* Frame 1:
- Change since task initialization: [...]
- What remains to task completion: [...]
- Evaluate task progress: [a value between 0.0 and 1.0]
* Frame 2: ...
* Frame 3: ...
(2) For Video clip B: What are the differences between each frame of video clip B and the task initialization image and the task completion image?
* Frame 1:
- Change since task initialization: [...]
- What remains to task completion: [...]
- Evaluate task progress: [a value between 0.0 and 1.0]
* Frame 2: ...
* Frame 3: ...
(3) For both Video clips: Compare the task completion progress in each frame of video A and B:
* Frame 1: (Which is closer to the completion image, and why)
* Frame 2: ...
* Frame 3: ...

- - - - - - - - - - - - - - - - - - - - - - - - - - - - - - - - - - - - - - - - - - - - - - - - - - - - - - - - - - - - - - - - - - - - -

**Labeling Template:**

You are a helpful assistant.
You are comparing two video clips A and B, showing partial execution of the following task: `<Task Description>`.
You are given the two video clips A and B analysis: `<Analyses from previous response>`
Your goal is to determine **which video better completes the task** and estimate how close each frame is to the completion state.
Please strictly follow the rules:
1. The video that is closer to the task completion image in its final frame is preferred.
2. Do NOT prefer a video just because it starts from the initialization image.
3. Ignore whether the video starts correctly - we only care how far it progresses.
4. Completion scores range from 0.0 (looks like init) to 1.0 (looks like completion). Use intermediate scores if necessary.
If both videos are similarly incomplete, you may return -1.
Format your output exactly like this:
#decision#: A or B or -1 #evaluate task completion for each frame in the video clip#:
A: [score1, score2, score3]
B: [score1, score2, score3]

*Table 13.* The prompt template for the preference label of CoRe.

---

**Prompt Template for Formal Reward Generation of CoRe**

---

**Initial System:** You are a reward engineer trying to write reward functions to solve reinforcement learning tasks as effective as possible. Your goal is to write a reward function for the environment that will help the agent learn the task described in text. The reward function signature is:

```python
def reward_function(observation, action, target_pos):
    ...
    return reward, {}
```

Your reward function should use variables defined in the above reward function signature as inputs.
The output of the reward function should consist of two items:
(1) the total reward,
(2) a dictionary of each individual reward component.
The total reward should be constrained to [-1, 1].
The total reward should be consistent with each individual reward component, which means that the sum of the individual reward component is equal to the total reward.
The code output should be formatted as a python code string:
""'python ... "''.
Some helpful tips for writing the reward function code:
(1) You may find it helpful to normalize the reward to a fixed range by applying transformations like np.exp to the overall reward or its components
(2) If you choose to transform a reward component, then you must also introduce a temperature parameter inside the transformation function; this parameter must be a named variable in the reward function and it must not be an input variable. Each transformed reward component should have its own temperature variable
(3) Most importantly, the reward code's input variables must only be variables defined in the reward function signature. Under no circumstance can you introduce new input variables

**User System:**

The Python environment is <Reward Info>. Write a reward function for the following task:<Task Description>.

**Policy Feedback:**

# Current Best Reward Function #
We trained a RL policy using the provided reward function code and tracked the values of the individual components in the reward function as well as global policy metrics such as task scores after every $episode_{freq}$ episodes and the maximum, mean, minimum values encountered:
# Performance metrics recorded during training, such as task progress and the evolution of reward components.#
Please carefully analyze the policy feedback and provide a new, improved reward function that can better solve the task. Some helpful tips for analyzing the policy feedback:
(1) If the task scores are always near zero, then you must rewrite the entire reward function
(2) If the values for a certain reward component are near identical throughout, then this means RL is not able to optimize this component as it is written. You may consider
(a) Changing its scale or the value of its temperature parameter
(b) Re-writing the reward component
(c) Discarding the reward component
(3) If some reward components' magnitude is significantly larger, then you must re-scale its value to a proper range Please analyze each existing reward component in the suggested manner above first, and then write the reward function code.
The output of the reward function should consist of two items:
(1) the total reward,
(2) a dictionary of each individual reward component.
The total reward should be constrained to [-1, 1].
The total reward should be consistent with each individual reward component, which means that the sum of the individual reward component is equal to the total reward.
The code output should be formatted as a python code string:
""'python ... "''.
Some helpful tips for writing the reward function code:
(1) You may find it helpful to normalize the reward to a fixed range by applying transformations like np.exp to the overall reward or its components
(2) If you choose to transform a reward component, then you must also introduce a temperature parameter inside the transformation function; this parameter must be a named variable in the reward function and it must not be an input variable. Each transformed reward component should have its own temperature variable
(3) Most importantly, the reward code's input variables must only be variables defined in the reward function signature. Under no circumstance can you introduce new input variables

*Table 14.* The prompt template for formal reward generation of CoRe.

**Prompt Template for Reward Info**

**MetaWorld Tasks:**

```
# The observation space is represented as a 6-tuple of the 3D Cartesian positions of
↪   the end-effector, a normalized measurement of how open the gripper is, the 3D
↪   position of the first object, the quaternion of the first object.
# In this task, first_obj_pos indicates the current position of the cube and
↪   first_obj_quat indicates the quaternion of the cube.
observation = np.concatenate(pos_hand[:3], gripper_distance_apart[0],
                             first_obj_pos[:3], first_obj_quat[:4],
                             )
# The action space is a 2-tuple consisting of the change in 3D space of the
↪   end-effector followed by a normalized torque that the gripper fingers should
↪   apply. The action range is [-1, 1].
action = [delta_x, delta_y, delta_z, gripper_torque]
# The target position is the target position of the cube
target_pos = self._target_pos
```

**Fold Cloh:**

```
# The observation space is represented as the 3D Cartesian coordinates of particles
↪   that make up a flexible object.
# In this task, top_left_posm, top_right_pos, bottom_left_pos, and bottom_right_pos
↪   respectively represent the current positions of the top left corner, top right
↪   corner, bottom left corner, and bottom right corner of the cloth.
# The cloth is laid flat in the horizontal plane represented by the X-axis and the
↪   Z-axis.
top_left_pos, top_right_pos, bottom_left_pos, bottom_right_pos = observation[:3],
↪   observation[15:18], observation[90:93], observation[105:108]
# The action space is the horizontal movement space in the top left corner of the
↪   cloth. The action range is [-0.15, 0.15].
action = [x_pos, z_pos]
```

**Straighten Rope:**

```
# The observation space is represented as the 3D Cartesian coordinates of particles
↪   that make up a flexible object.
# In this task, end_point1, end_point2 respectively represent the current positions
↪   of the two endpoints of the rope.
# The rope is laid flat in the horizontal plane represented by the X-axis and the
↪   Z-axis.
end_point1, end_point2 = observation[:3], observation[27:30]
# The action space is the changes in 3D space of the two endpoints of the rope. The
↪   action range is [-0.01, 0.01].
end_point1_delta_x, end_point1_delta_y, end_point1_delta_z = action[:3]
end_point1_delta_x, end_point1_delta_y, end_point1_delta_z = action[4:7]
# Please note that target_pos is not required for this task, and action can be used
↪   or not.
```

**Pass Water:**

```
# In this task, cup_pos_x represent the current position of the cup along X-axis.
↪   cup_length, cup_width, cup_height respectively represent the size of the cup.
# water_height represents the height of the water in the cup. water_in_cup,
↪   water_out_of_cup respectively represent the proportion of water in and out of the
↪   cup.
# The cup is laid in the horizontal plane represented by the X-axis and the Z-axis.
observation = [cup_pos_x, cup_length, cup_width, cup_height, water_height,
↪   water_in_cup, water_out_of_cup]
# The action space is the changes of the cup along the X-axis. The action range is
↪   [-0.011, 0.011]. Please note that action can be used or not.
action = [cup_pox_x_movement]
# target_pos is the target position that the cup needs to reach.
```

*Table 15.* The prompt template for reward info of CoRe.

