# OpenReview forum: "CoRe: Combined Rewards with Vision-Language Model Feedback for Preference-Aligned Reinforcement Learning"
_ICML.cc/2026/Conference — ICML 2026 regular_

### Official Review · Reviewer_j1dQ · 2026-02-13

**Soundness:** 4
**Presentation:** 2
**Significance:** 3
**Originality:** 3
**Overall Recommendation:** 4
**Confidence:** 2

**Summary:**

The paper proposes CoRe, a framework that automates reward design in reinforcement learning by decomposing the reward function into two complementary components: Formal Rewards (FR) and Residual Rewards (RR). FRs are code-based functions generated by LLMs based on task descriptions and iteratively refined via preference feedback, capturing explicit task structures. RRs are neural network-based rewards learned from video-level preferences provided by Vision-Language Models (VLMs), capturing implicit and nuanced task information. The authors introduce a Formal Reward Module (FRM) and a Residual Reward Module (RRM) that work synergistically.

**Compliance With Llm Reviewing Policy:**

Affirmed.

**Final Justification:**

The authors have addressed my concerns.

**Key Questions For Authors:**

1.  Can you provide a detailed breakdown of the computational cost (token usage, API costs) and training time (wall-clock) compared to baselines like Eureka? Does the frequent querying of VLMs for the RRM introduce a bottleneck?
2.  The method uses state-of-the-art proprietary models (Gemini 2.0 Flash). How does CoRe perform if the VLM is replaced with an open-source alternative like LLaVA-Video or a smaller model?
3.  In cases where the Formal Reward (code) and Residual Reward (VLM preference) provide strongly conflicting signals (e.g., code rewards a safety violation that the VLM ignores, or vice versa), how does the additive formulation ($L_{Reward} + \alpha L_{KL}$) handle this? Did you observe any instability in such scenarios?

**Limitations:**

1.The authors should explicitly discuss the financial and time costs associated with the API calls required for their method, as this impacts scalability.
2.The evaluation primarily focuses on structured tabletop manipulation tasks (MetaWorld, SoftGym) and controlled real-world scenarios. These environments do not fully represent the complexity of unstructured real-world settings, where VLM-based perception and reward generation might be significantly less reliable.

**Strengths And Weaknesses:**

**Strengths:**
1.  Overall, an important concept presented by the manuscript is the decomposition of the reward signal into "Formal" (explicit, code-based) and "Residual" (implicit, neural-based) components. This hybrid approach effectively addresses the limitations of relying solely on LLM-generated code (which can be sparse or miss nuances) or VLM-learned rewards (which can be noisy or data-hungry). This design is cognitively inspired and technically sound.
2.  The experimental evaluation is comprehensive. CoRe demonstrates superior performance on both rigid body (MetaWorld) and deformable object (SoftGym) tasks, significantly outperforming strong baselines like Eureka and PrefVLM. The inclusion of real-world robot experiments (UR5) adds significant weight to the claims of robustness and transferability.

**Weaknesses:**
1.  The proposed framework relies heavily on querying large proprietary models (GPT-4 for code, Gemini/VLM for preferences) during the training loop. The paper lacks a detailed analysis of the wall-clock time, token cost, and latency introduced by these queries compared to standard RL or simpler baselines.

---

> ### Author Rebuttal · Authors · 2026-03-31
>
> Thank you for evaluating our design and experiments positively.
>
> W1: We provide a detailed table of token usage, API cost, and wall-clock time for all methods.
> Despite relying on LLM/VLM queries, CoRe remains efficient: it uses ~2.00M tokens, incurs ~0.37$ API cost, and completes training in 2.15h, achieving lower cost, better performance, and faster runtime. Methods such as RL-VLM-F demand far more tokens and longer time. Notably, model query latency does not dominate training, since CoRe reduces query frequency via the structured FRM and limited preference queries. CoRe introduces moderate overhead over standard RL but achieves a cost–runtime–performance trade-off.
> |Method|Token(M)|API(\$)|Time(h)|
> |:----------|:------:|:-----:|:-----:|
> |SAC|—|—|0.97|
> |CLIP Score|—|—|2.58|
> |Eureka|0.03|0.03|1.87|
> |Text2Reward|0.03|0.02|5.73|
> |RL-VLM-F|5.50|0.79|6.72|
> |PrefVLM|—|—|1.82|
> |ERL-VLM|3.19|0.55|5.80|
> |CoRe|2.00|0.37|2.15|
>
> Q1: We provide a detailed breakdown in W1 Table. CoRe uses ~2.00M tokens, with an cost of ~0.37 USD and 2.15h wall-clock time, which is comparable to Eureka (0.03M tokens, ~0.03 USD, 1.87h). Although Eureka is cheaper, its performance is lower than CoRe. Compared with VLM-based methods, CoRe achieves much higher efficiency in token usage, runtime, and performance. VLM queries are not a bottleneck: in CoRe, queries are sparse and periodic.  Moreover, FRM provides a strong stable prior, reducing the demand for frequent RRM feedback.
>
> Q2: We evaluate CoRe using open-source VLMs and report the results.
> Replacing Gemini 2.0 Flash with Qwen3-VL-8B-Instruct still yields strong performance (84.0% in Sweep Into, 88.7% in Hammer), verifying that CoRe does not depend on specific proprietary models.
> Video-LLaVA-7B exhibits insufficient instruction-following and video understanding to generate reliable preference labels, resulting in ineffective training.
> These results confirm that CoRe is model-agnostic and performs well with open-source VLMs. As open-source VLMs continue to advance, we expect CoRe to achieve comparable performance.
> |Method|Sweep Into|Hammer|
> |:-------------------|:--------:|:----:|
> |Video-LLaVa-7B|—|—|
> |Qwen3-VL-8B-Instruct|84.0%|88.7%|
> |Gemini 2.0 Flash|97.3%|98.0%|
>
> Q3: Conflicts between FRM and RRM may arise in early training when preference feedback is limited. This is observed in Peg Insert (Fig. 3g), where performance is initially unstable within the first 2x10^5 steps, likely due to conflicts, but improves rapidly once aligned. CoRe has the ability to resolve such conflicts through iterative preference alignment. FRM is iteratively optimized through reward–preference alignment: generated reward codes are evaluated against VLM-labeled preferences, and misaligned rewards are discarded and regenerated. RRM is directly learned from preferences, so both modules are consistently guided toward human preferences. As training proceeds and feedback accumulates, FRM and RRM gradually align, and conflicts diminish. The iterative alignment mechanism detects and corrects conflicting signals, ensuring convergence to consistent rewards.
>
> L1: API calls’ financial and time costs are critical for scalability and be discussed in the revision.
> In CoRe, FRM has low query frequency, while RRM relies on continuous VLM-based preference labeling. Although our design reduces the number of queries, the overall cost is still affected by API pricing and network latency, which may limit large-scale scalability. In our experiments, one full CoRe run uses ~2.00M tokens, costs ~0.37$, and takes 2.15h of training time (table in W1).
> We also clarify that CoRe can be deployed with open-source models, which eliminates API expenses and reduces latency, offering a more scalable solution.
>
> L2: We recognize that broader evaluation in unstructured settings is needed and discuss this in the revision.
> Moving from structured tabletop to unstructured real-world environments poses challenges for VLM reliability. For fair comparison and controllability, our evaluation mainly uses standardized benchmarks (MetaWorld, SoftGym) to rigorously validate all methods, though they cannot fully represent real-world complexity. CoRe is also tested on noisy, less controlled real-world UR5 tasks and achieves better performance.
> CoRe is inherently more robust than pure VLM-based methods: FRM provides a structured backbone that preserves task-oriented behavior even when VLM perception fails as noisy inputs.
> Additional disturbance experiments with visual noise (a randomly moving red cube) in Sweep Into and Hammer show minor performance drops for CoRe, confirming its robustness to perceptual noise.
> We acknowledge that a more comprehensive evaluation in unstructured environments is essential for future work. Moving forward, we aim to further refine our approach to enhance its robustness and generalizability across a wider range of complex scenarios.
> |Method|Sweep Into|Hammer|
> |:-----|:--------:|:----:|
> |Clear|97.3%|98%|
> |Noisy|91.7%|96%|

---

> > ### Author Rebuttal · Reviewer_j1dQ · 2026-04-03
> >
> > The authors' response has addressed most of my concerns. I will maintain my current score.

---

> > > ### Author Response · Authors · 2026-04-04
> > >
> > > Thank you for your thoughtful review of our rebuttal. We are glad to see that our response addressed your concerns. We sincerely appreciate your comments, which have helped improve the evaluation of our work.

---

### Official Review · Reviewer_sgya · 2026-02-28

**Soundness:** 3
**Presentation:** 2
**Significance:** 2
**Originality:** 2
**Overall Recommendation:** 4
**Confidence:** 3

**Summary:**

CoRe (Combined Rewards) is a hybrid reward design framework that enables preference-aligned reinforcement learning without human supervision by integrating vision-language models (VLMs). It decomposes rewards into Formal Rewards (FR), which are automatically generated and refined executable reward functions based on structured task knowledge, and Residual Rewards (RR), which learn implicit human preferences from observational data. The Formal Reward Module (FRM) improves reward reliability through reward-preference alignment using VLM-generated labels, reducing reward hacking, while the Residual Reward Module (RRM) learns fine-grained nonlinear rewards from video-level preferences to capture complex task dynamics. Together, these modules form a synergistic optimization loop that enables efficient, automated reward construction, leading to state-of-the-art performance across multiple simulated and real-world robotic manipulation tasks.

**Compliance With Llm Reviewing Policy:**

Affirmed.

**Final Justification:**

The authors' response has addressed most of my concerns.

**Key Questions For Authors:**

- Q1: The method section is currently difficult to follow due to its presentation and organization. Could the authors consider reorganizing the method description to provide a clearer and more structured end-to-end explanation of the proposed framework?
- Q2: Can the authors clarify the meaning of different arrows and data flows in the main method figure? A step-by-step description aligned with the diagram would help readers better understand the overall pipeline.
- Q3: Would it be possible to include a more explicit algorithm or pseudo-code summarizing the full CoRe training pipeline to improve reproducibility and readability?

**Limitations:**

yes

**Strengths And Weaknesses:**

## Strengths
- Soundness: The method builds on established preference-based RL and introduces VLM-generated video-level preferences with importance weighting. The approach is technically reasonable and supported by comparisons with strong baselines.

- Significance: Removing human annotation is practically useful, but the paper provides limited analysis of broader applicability, which constrains its overall impact.

- Originality: The combination of VLM-based preferences and importance weighting offers some novelty, but overall the contribution is largely incremental.

## Weaknesses
- Presentation & Significance: Limited clarity in method presentation. The overall methodological description is difficult to follow, and the interactions between different components of the framework are not always clearly explained.

- Presentation: Confusing visualization in the main method figure. The method diagram contains multiple arrows and information flows that are hard to interpret, making it challenging to understand how different modules interact during training.

- Presentation:  Insufficient explanation of equations and figures. Several introduced equations are not adequately explained or explicitly linked to the corresponding figures and captions, which reduces accessibility for readers trying to understand the implementation details.

---

> ### Author Rebuttal · Authors · 2026-03-31
>
> We appreciate your constructive feedback and address your concerns regarding clarity, presentation, and organization below.
>
> W1&Q1: We reorganize and revise the method section for a clearer, structured, end-to-end presentation.
> CoRe decomposes the task reward into two complementary components: **Formal Reward (FR)** to model structured, explicit task objectives, and **Residual Reward (RR)** to capture implicit, hard-to-specify preferences from visual observations.
> Accordingly, CoRe contains two parallel and interactive modules:
>
> **Formal Reward Module (FRM)** uses LLMs to generate and iteratively refine code-based formal rewards guided by task knowledge and preference feedback.
>
> **Residual Reward Module (RRM)** learns a residual reward  from VLM-labeled video-level preferences and frame-wise importance weights, capturing fine-grained and perceptual aspects.
>
> **The overall pipeline is as follows:**
> During RL training, collected trajectories are segmented into clips and fed to the VLM for preference labels and importance weights. RRM is updated with these signals. In parallel, FRM performs reward–preference alignment with the same preference labels to iteratively improve formal reward code.
> The final reward for policy optimization is the sum of FR and RR, benefiting from structured guidance and flexible preference alignment.
>
> W2&Q2: We clarify the modules and data flows as follows:
>
> **Module overview:**
> The blue module (top) is the Formal Reward Module (FRM), which generates and iteratively refines code-based rewards using LLMs and preference alignment.
> The purple module (bottom) is the Residual Reward Module (RRM), which learns residual rewards from VLM-based video preferences and importance weights.
> The white module (right) is the RL interaction module, where the robot interacts with the environment to collect trajectories and update the policy.
>
> **Arrow semantics:**
>
> 1. Solid black arrows: data flow (e.g., trajectories, preference labels, buffers).
> 2. Dashed black arrows: reward learning/updates.
> 3. Bold arrows: LLM/VLM I/O (e.g., natural language, video frames, reward code).
>
> **Step-by-step pipeline:**
>
> 1. The RL module interacts with the environment to collect trajectories.
> 2. In RRM (bottom), trajectories are sampled into segment pairs, combined with task context, and sent to the VLM to obtain video-level preferences and importance weights, which are stored in the preference buffer and used to update the residual reward model.
> 3. In parallel, FRM (top) uses LLMs to generate candidate reward codes based on task knowledge and policy feedback, selecting  the best one via reward–preference alignment using the same preference buffer.
> 4. The final reward (FR + RR) is used to relabel experience data and optimize the policy.
>
> We further revise the figure and caption to clearly reflect these roles and interactions, improving alignment between the diagram and method description.
>
> W3: We have revised the paper to provide clearer explanations of key equations.
>
> **Eq. (2)–(3) (FRM: Reward–Preference Alignment):** These equations define how candidate formal rewards are evaluated. Each reward is scored by its consistency with preference labels in the dataset. Eq. (3) computes the preference prediction accuracy of a reward over trajectory pairs, and Eq. (2) selects the reward that best aligns with preferences. This corresponds to Fig. 2(b), where reward selection is guided by preference consistency.
>
> **Eq. (4) (RRM: Preference labeling):** This equation formalizes how the VLM generates video-level preference labels from task description, start/goal images, and trajectory videos.
>
> **Eq. (5)–(6) (RRM: Reward learning objective):** Eq. (5) defines the overall loss for the residual reward model, combining:
> 	(1) a preference loss to match predicted rewards with preference labels, and
> 	(2) a KL regularization term (Eq. (6)) that aligns predicted per-frame rewards with VLM-estimated importance weights, improving credit assignment and stabilizing learning.
>
> We further connect these equations to Fig. 1 by explicitly indicating how they correspond to FRM optimization and RRM learning, ensuring a clearer mapping between equations, figures, and implementation.
>
>
>
> Q3: We note that a full pseudo-code of the CoRe training pipeline is provided in the Appendix (Algorithm 1). We make this more prominent in the main text and add a brief summary of CoRe's iterative pipeline to improve readability.
>
> (1) initialize policy, reward models, and buffers;
>
> (2) collect trajectories via environment interaction;
>
> (3) periodically sample segment pairs and query the VLM to obtain preference labels and importance weights, which are used to update RRM;
>
> (4) in parallel, generate candidate formal rewards via LLM and select the best one using reward–preference alignment;
>
> (5) combine FRM and RRM rewards to relabel data and update the policy;
>
> (6) repeat the above steps until convergence.

---

> > ### Author Rebuttal · Reviewer_sgya · 2026-04-04
> >
> > The authors' response has addressed most of my concerns. I will raise my score.

---

> > > ### Author Response · Authors · 2026-04-04
> > >
> > > We appreciate you taking the time to go through our responses. It is good to know that our response resolved your concerns, and we are grateful for the increased score. Your suggestions have been helpful in improving the presentation and clarity of the manuscript.

---

### Official Review · Reviewer_a4gL · 2026-03-12

**Soundness:** 3
**Presentation:** 3
**Significance:** 2
**Originality:** 2
**Overall Recommendation:** 4
**Confidence:** 3

**Summary:**

This paper proposes CoRe, a framework for preference-aligned reinforcement learning that decomposes the reward into a formal reward and a residual reward. The formal reward is generated and refined as code by an LLM, while the residual reward is learned from VLM-generated video preference labels and frame-importance signals. Experiments on simulated and real-world robotic manipulation tasks show that the method outperforms several existing LLM/VLM-based reward learning baselines.

**Compliance With Llm Reviewing Policy:**

Affirmed.

**Final Justification:**

The rebuttal addressed most of my concerns, particularly regarding the temporal behavior of the residual reward model. I remain somewhat concerned about the comparability of video segments from different task stages, although the clarification about excluding incomparable pairs partially alleviates this issue. Overall, I find the method technically sound and reasonably well motivated, and I maintain my score of 4.

**Key Questions For Authors:**

Q1.
How do you compare randomly sampled video segments that come from different stages of the same task? For example, how do you ensure that a segment showing approach behavior and a segment showing task completion are meaningfully comparable for preference labeling?

Q2.
Can your method distinguish event completion from state persistence? For example, how does the learned residual reward avoid repeatedly rewarding a visually salient state that should only be rewarded as a transient event?

Q3.
The method relies on commercial foundation models, specifically GPT-4.1 mini for FRM and Gemini 2.0 Flash for RRM. What is the approximate API cost of one full experiment? Also, is there any particular reason for choosing GPT for FRM and Gemini for RRM, rather than using the same model family for both modules?

**Limitations:**

The authors do not explicitly discuss the limitations of their work. Specifically, the lack of explicit bounds on the uncertainty of the LLM/VLM outputs could introduce hidden risks.

**Strengths And Weaknesses:**

## Strengths

S1. Good motivation.
The paper is well motivated. It addresses an important problem in reinforcement learning for robotics: hand-designed rewards are hard to specify, while preference-based reward learning is often expensive and unstable.

S2. Well-designed framework.
The overall framework is thoughtfully designed. In particular, the idea of combining a structured formal reward with a learned residual reward is intuitive, and the use of frame-importance signals as auxiliary supervision for the residual reward model is interesting.

S3. Good writing.
The paper is clearly written and generally easy to follow.

## Weaknesses

W1. The comparability of video preference labels is questionable.
The paper assumes that randomly sampled video segments can be meaningfully compared. However, this assumption is unclear when the two segments come from different stages of a task. For example, one segment may only show the robot moving from the initial state toward a drawer, while another segment may show the robot already opening the drawer. In this case, it is not obvious that these two segments are directly comparable, either in terms of preference or in terms of frame importance. This raises concerns about the validity of the preference labels used to train the residual reward model.

W2. The residual reward model is frame-based and does not explicitly capture temporal information.
The residual reward model appears to operate on single frames rather than short temporal windows, which makes it difficult to distinguish between transient task events and persistent visual states. As a result, it is not clear that the RRM will always contribute positively. For example, in a drawer-opening task, grasping the handle may correspond to an important frame and thus receive a relatively high reward. However, if the learned residual reward overvalues that visual state, the policy may prefer maintaining the grasped state instead of actually pulling the drawer open. More generally, the presence of the RRM may bias the overall reward optimization in undesirable ways, which seems to be a potential weakness of the method.

---

> ### Author Rebuttal · Authors · 2026-03-31
>
> Thank you for the positive feedback on our motivation, design, and clarity.
>
> W1: The concern is valid when segments are not directly comparable.
> For segments from different task stages, preference is not based on state but on overall task progress and execution quality. For example, a segment showing *opening the drawer* is preferred only if it is executed well and reflects higher task completion; otherwise, a well-executed *approach* segment can be preferred. In this sense, preferences reflect a holistic assessment of progress and quality, rather than strict stage-wise alignment.
> In some special cases, the VLM is also allowed to output an “incomparable” label, and such pairs are excluded from RRM training, preventing invalid supervision.
> This design ensures that only meaningful and consistent preferences are used for training, mitigating issues from cross-stage comparisons.
>
> W2: RRM performs frame-level inference for efficiency, but its learning is not frame-independent.
> RRM is trained from video-level preferences, which inherently encode temporal dynamics. Compared to single-frame, such preferences better capture sequential intent (e.g., *opening the drawer* is preferred over *holding the handle*). In addition, we introduce importance weights that reflect relative progress within a trajectory, assigning higher value to later, task-advancing states.
> As a result, RRM does not overvalue transient states (e.g., grasping), but instead learns a reward aligned with task progress. This is evidenced in Fig. 5, where the learned reward evolves smoothly and monotonically with task progress, without plateaus or spikes at intermediate states.
>
> Q1: When comparisons are made across stages, preferences are determined by overall task progress and execution quality, rather than strict stage alignment. A later-stage segment (e.g., task completion) is preferred only if it is well-executed; otherwise, an earlier-stage but better-executed segment can be preferred.
> Furthermore, the VLM is allowed to output an “incomparable” label when two segments are not meaningfully comparable; such pairs are excluded from RRM training, avoiding invalid supervision.
> Thus, CoRe ensures that only meaningful and consistent comparisons contribute to training, mitigating issues from randomly sampled cross-stage segments.
>
> Q2: RRM performs frame-level inference for efficiency, but its learning captures temporal structure through video-level preferences.
> Specifically, RRM is trained from video-level comparisons, which encode sequential intent (e.g., opening the drawer is preferred over holding the handle). Combined with importance weighting that emphasizes later, task-advancing states, the learned reward aligns with task progression rather than static visual saliency. As shown in Fig. 5, the reward evolves smoothly and monotonically, without overvaluing intermediate states.
> In addition, FRM provides explicit physical grounding (e.g., drawer displacement). If the agent remains in a persistent state (e.g., only grasping), FRM reward stops increasing. To maximize return, the policy is therefore driven to complete the transient action (e.g., pulling), rather than exploiting a static state.
> This design enables CoRe to distinguish event completion from state persistence and avoid repeated reward on transient states.
>
> Q3: We provide a detailed cost across all methods. A full CoRe experiment uses ~2.00M tokens, corresponding to an API cost of ~0.37\$ and 2.15 hours of training.
> Regarding model choices, we select different models for FRM and RRM based on their functional strengths and cost efficiency. FRM relies on code generation, where GPT models are more reliable for structured reasoning and code synthesis. In contrast, RRM requires large-scale visual preference queries; Gemini offers strong visual understanding at lower cost, making it more suitable. We believe that with the continuous advancement of foundation models and their improving capabilities in code generation and video understanding, CoRe can eventually be unified into a single model family in the future.
> |Method|Token(M)|API(\$)|Time(h)|
> |:----------|:------:|:-----:|:-----:|
> |SAC|—|—|0.97|
> |CLIP Score|—|—|2.58|
> |Eureka|0.03|0.03|1.87|
> |Text2Reward|0.03|0.02|5.73|
> |RL-VLM-F|5.50|0.79|6.72|
> |PrefVLM|—|—|1.82|
> |ERL-VLM|3.19|0.55|5.80|
> |CoRe|2.00|0.37|2.15|
>
> L1: We agree with the reviewer that this is an important limitation.
> While CoRe mitigates uncertainty through structured FRM design, aggregated preference learning, and iterative alignment, the outputs of LLM/VLMs still lack explicit uncertainty bounds, which may introduce hidden risks in certain scenarios. In addition, the framework incurs non-negligible training cost due to iterative querying.
> We explicitly discuss these limitations in the revised paper and clarify potential risks and future directions (e.g., uncertainty-aware modeling and more efficient querying strategies).

---

> > ### Author Rebuttal · Reviewer_a4gL · 2026-04-03
> >
> > Thank you for the detailed rebuttal. The responses addressed most of my concerns, especially regarding the residual reward design, and made me more positive about the paper overall. I believe the current score of 4 remains appropriate.

---

> > > ### Author Response · Authors · 2026-04-03
> > >
> > > Thank you for your thoughtful feedback and for taking the time to carefully review our rebuttal. We are glad that our responses helped address your concerns. We appreciate your positive assessment of our work and your constructive comments, which have been very helpful in improving the paper.

---

### Official Review · Reviewer_UWrA · 2026-03-13

**Soundness:** 2
**Presentation:** 3
**Significance:** 2
**Originality:** 2
**Overall Recommendation:** 4
**Confidence:** 4

**Summary:**

This paper proposed a hybrid framework to automatically construct reward function for preference alignment with LLM/VLM. It includes two primaryh modules:
1. Formal reward module (FRM) use LLM to generate code-based rewards that are relevant to task knowledge.
2. Residual reward module (RRM) learns a residual reward model from video-level preference labels by VLM to capture nuanced rewards that complement FR for enhanced alignment.

The method is validated via a set of simulated environments in MetaWorld and SoftGym. The results and ablations suggest that the proposed method CoRe achieves the highest correlation with actual task progress and can effectively improve learning efficiency.

**Compliance With Llm Reviewing Policy:**

Affirmed.

**Final Justification:**

The authors have addressed my concerns.

**Key Questions For Authors:**

I've raised some questions in the weakness part. In addition
1. As FRM requires explicit state definitions, how does CoRe differ from traditional reward engineering beyond using an LLM to write the Python code?
2. How does the system prevent noise introduced by limited feedback, especially in more complex, unconstrained environments?
3. To what extent the performance of CoRe depends on the specific LLM/VLM prompts?

**Limitations:**

No limitation has been discussed. A thorough analysis of the aforementioned weakness will be appreciated.

**Strengths And Weaknesses:**

Strength
1. The idea of decomposing reward into a formal reward and a residual reward is intuitive and straightforward. Based on the results this idea also seems effective.
2. Robust reward generation without manual reward engineering is very practice for real-world RL application.
3. The paper delivery is clear and easy to follow.

Weakness
1. The primary limitation of the proposed method is it's heavy reliance on the LLM/VLM models, which pretty much means the performance is upper bounded by these models. With growing capability of image/video understanding of those foundation models, such reward decomposition might not be necessary.
2. Due to the reliance on LLM/VLM, the authors mentioned that the FRM is highly sensitive to label noise, which cannot be easily eliminated given the fact that those labels rely entirely on the VLM's capability.
3. FRM still requires "task knowledge", which need manual engineering.
4. The proposed framework involves multiple iterative loops and repeated calling to LLM/VLM, what are the costs for computation and inference?
5. The proposed method is only validated on two simple simulated environments with clear background and highly controlled states, does it still effective in real-world scenarios with noisy and distractive surroundings and backgrounds?

---

> ### Author Rebuttal · Authors · 2026-03-31
>
> We appreciate the constructive feedback and the recognition of our method.
>
> W1:  We agree stronger LLM/VLMs improve reward generation. However, our goal is not to compensate for model limitations, but to address inherent limits of single-form rewards. CoRe decomposes rewards into formal and residual components to leverage their strengths.
> We validate this in Hammer with advanced models. Stronger models with single-form rewards yield limited gains but much higher cost (FRM: 54.0% → 54.7% with ~9× cost; RRM: 49.3% → 62.6% with ~6× cost). This show scaling model does not resolve the limitations of single-form rewards. CoRe achieves better performance–efficiency trade-offs, highlighting the necessity of decomposition.
> |Method|Success Rate(%)|Cost(\$)|
> |:--------------------------------------|:-------------:|:------:|
> |FRM with GPT-4.1 mini|54.0|0.06|
> |FRM with GPT-5.4|54.7|0.53|
> |RRM with Gemini 2.0 Flash|49.3|0.31|
> |RRM with Gemini 3 Flash|62.6|1.88|
> |CoRe with GPT-4.1 mini&Gemini 2.0 Flash|97.3|0.37|
>
> W2: Our method’s overall design ensures robustness. FRM leverages the entire preference dataset rather than individual labels, reducing sensitivity to noisy annotations. RRM similarly captures consistent patterns from large-scale preferences. Empirically, CoRe achieves a 99.0% average success rate on MetaWorld (Table 1) and remains robust in real-world (Table 4).
>
> W3: Task knowledge is primarily provided by the environment and involves minimal overhead. Only a one-time declarative task description is required  without any design and tuning. By assigning complex details to RRM, FRM only needs basic variable definitions. Besides, such task knowledge is unavoidable: any LLM needs variable semantics to generate code.
>
> W4: We provide detailed cost breakdowns for all methods. Despite involving LLM/VLM calls, CoRe remains computationally and cost-efficient: it uses \~2.00M tokens and incurs lower API cost (~0.37\$), and maintaining a short training time (2.15h). Methods like RL-VLM-F require more cost. CoRe achieves a trade-off between low cost, fast training, and strong performance.
> |Method|Token(M)|API(\$)|Time(h)|
> |:----------|:------:|:-----:|:-----:|
> |SAC|—|—|0.97|
> |CLIP Score|—|—|2.58|
> |Eureka|0.03|0.03|1.87|
> |Text2Reward|0.03|0.02|5.73|
> |RL-VLM-F|5.50|0.79|6.72|
> |PrefVLM|—|—|1.82|
> |ERL-VLM|3.19|0.55|5.80|
> |CoRe|2.00|0.37|2.15|
>
> W5: The simulated environment used in this work is a widely adopted benchmark in recent RL research [1] [2]. serving as a standard testbed to ensure fair comparisons.
> We further evaluate CoRe under distracting settings: we introduce visual disturbances (a randomly moving red cube) in Sweep Into and Hammer, and CoRe maintains stable performance with minor degradation.
> As reported in Sec. 4.4, CoRe is validated on real-world with unpredictable noise, outperforming baselines, and its performance degradation during sim-to-real transfer is minimal (Table 4).
> These confirm that CoRe generalizes beyond controlled simulations and remains effective in real-world environments.
> |Method|Sweep Into|Hammer|
> |:-----|:--------:|:----:|
> |Clear|97.3%|98%|
> |Noisy|91.7%|96%|
> [1]: Rl-vlm-f: Reinforcement learning from vision language foundation model feedback. ICML. 2024.
> [2]: Ehancing rating-based reinforcement learning to effectively leverage feedback from large vision-language models. ICML. 2025.
>
> Q1: CoRe is a hybrid, automated reward learning framework:
> 1. It only requires high-level task knowledge, with the LLM automatically handling reward formulation, eliminating hand-crafted shaping functions and weight tuning.
> 2. It uses a preference-driven to automatically refine rewards, which removes manual iteration and debugging to fix code.
> 3. Its FRM captures explicit task structure, while the RRM learns implicit, nuanced behaviors from visual data, without require explicit coding of all details.
>
> Q2: CoRe mitigates noise from limited feedback through hybrid reward and preference aggregation:
> 1. FRM provides a structured signal, ensuring stable learning even when RRM feedback is sparse or noisy.
> 2. RRM are learned via two-stage multimodal queries with importance weighting, improving feedback. RRM is trained on aggregated preferences, where averaging effects and importance prevent overfitting and yield smoother rewards (Fig. 5).
> 3. CoRe continuously and iteratively refines the combined reward.
> Real-world performance demonstrating robustness under limited and noisy feedback.
>
> Q3: CoRe is insensitive to prompt design. Across all experimental tasks, we use identical prompt templates, with only a natural language task description and knowledge (Appx. Tab. 9–10) and no careful prompt engineering.
> Prompts merely initialize learning, while final performance depends on iterative optimization: FRM rewards are refined via feedback loops (not expected to be perfect from a single prompt); RRM adopts a two-stage CoT prompt, requiring explicit frame-wise reasoning before preference prediction to stabilize outputs.

---

> > ### Author Rebuttal · Reviewer_UWrA · 2026-04-04
> >
> > The authors have addressed most of my concerns. I'll raise my score accordingly.

---

> > > ### Author Response · Authors · 2026-04-04
> > >
> > > Thank you for your time and effort in reviewing our work. We are glad our rebuttal addressed your concerns, and we greatly appreciate the increased score. Your feedback has been very helpful in strengthening the paper, and we truly value your support.

---

### Decision · Program_Chairs · 2026-04-30

**Decision:**

Accept (regular)

**Comment:**

The paper proposes CoRe, a framework that automates reward design in reinforcement learning by decomposing the reward function into two complementary components: Formal Rewards (FR) and Residual Rewards (RR). FRs are code-based functions generated by LLMs based on task descriptions and iteratively refined via preference feedback, capturing explicit task structures. RRs are neural network-based rewards learned from video-level preferences provided by Vision-Language Models (VLMs), capturing implicit and nuanced task information. The authors introduce a Formal Reward Module (FRM) and a Residual Reward Module (RRM) that work synergistically. Experiments on simulated and real-world robotic manipulation tasks show that the method outperforms several existing LLM/VLM-based reward learning baselines.

The reviewers agree that the proposed idea to automate reward design is well-motivated, intuitive, effective (UWrA, a4gL) and useful for real-world RL application (UWrA). The paper is well-written and easy to follow (UWrA, a4gL). The proposed framework is cognitively inspired, well-designed  and technically sound (a4gL, sgya, j1dQ). The experimental evaluation is comprehensive (j1dQ) and supported by comparisons with strong baselines (sgya). Additionally, the inclusion of real-world robot experiments (UR5) adds significant weight to the claims of robustness and transferability (j1dQ).

However, the reviewers also note that the proposed framework relies heavily on querying large proprietary models (GPT-4 for code, Gemini/VLM for preferences) during the training loop (UWrA, j1dQ) and authors do not explicitly discuss the financial and time costs associated with the API calls required for their method, as this impacts scalability (j1dQ). The authors have provided this analysis during the rebuttal which successfully addressed the reviewer’s question, though it should be included in the revised paper. Additionally, the lack of explicit bounds on the uncertainty of the LLM/VLM outputs could introduce hidden risks (a4gL). Authors acknowledge the risks and explicitly discuss these limitations in the rebuttal and clarify potential risks and future directions (e.g., uncertainty-aware modeling and more efficient querying strategies). This discussion should also be added to the finalized version of the paper.

Moreover, the evaluation primarily focuses on structured tabletop manipulation tasks (MetaWorld, SoftGym) and controlled real-world scenarios. These environments do not fully represent the complexity of unstructured real-world settings, where VLM-based perception and reward generation might be significantly less reliable (j1dQ). The authors acknowledge this limitation and mention that this will be addressed as part of future work.

Two reviewers (UWrA, sgya) raised their score to 4 (weak accept) while the two other reviewers (a4gL, j1dQ) maintained their score of 4 after the rebuttal. In the final justification, reviewer a4gL mentions that they remain somewhat concerned about the comparability of video segments from different task stages, although the clarification about excluding incomparable pairs partially alleviates this issue. The authors addressed most concerns raised by each reviewer and overall, all reviewers find the method technically sound and reasonably well motivated. In support of all reviewers recommending the paper for weak acceptance, I recommend the paper for weak acceptance as well.